# Coupling of Slack and Na$_V$1.6 sensitizes Slack to quinidine blockade and guides anti-seizure strategy development

Tian Yuan[1†], Yifan Wang[1†], Yuchen Jin[1†], Hui Yang[1†], Shuai Xu[1], Heng Zhang[2], Qian Chen[1], Na Li[1], Xinyue Ma[1], Huifang Song[1], Chao Peng[1], Ze Geng[1], Jie Dong[1], Guifang Duan[1], Qi Sun[1], Yang Yang[3], Fan Yang[2,4], Zhuo Huang[1,5]*

[1]State Key Laboratory of Natural and Biomimetic Drugs, Department of Molecular and Cellular Pharmacology, School of Pharmaceutical Sciences, Peking University Health Science Center, Beijing, China; [2]NHC and CAMS Key Laboratory of Medical Neurobiology, MOE Frontier Science Center for Brain Research and Brain-Machine Integration, School of Brain Science and Brain Medicine, Zhejiang University, Zhejiang, China; [3]Department of Medicinal Chemistry and Molecular Pharmacology, College of Pharmacy, Purdue University, West Lafayette, United States; [4]Department of Biophysics, Kidney Disease Center of the First Affiliated Hospital, Zhejiang University School of Medicine, Hangzhou, Zhejiang, China; [5]IDG/McGovern Institute for Brain Research, Peking University, Beijing, China

*For correspondence:
huangz@hsc.pku.edu.cn

[†]These authors contributed equally to this work

**Abstract** Quinidine has been used as an anticonvulsant to treat patients with KCNT1-related epilepsy by targeting gain-of-function KCNT1 pathogenic mutant variants. However, the detailed mechanism underlying quinidine's blockade against KCNT1 (Slack) remains elusive. Here, we report a functional and physical coupling of the voltage-gated sodium channel Na$_V$1.6 and Slack. Na$_V$1.6 binds to and highly sensitizes Slack to quinidine blockade. Homozygous knockout of Na$_V$1.6 reduces the sensitivity of native sodium-activated potassium currents to quinidine blockade. Na$_V$1.6-mediated sensitization requires the involvement of Na$_V$1.6's N- and C-termini binding to Slack's C-terminus and is enhanced by transient sodium influx through Na$_V$1.6. Moreover, disrupting the Slack-Na$_V$1.6 interaction by viral expression of Slack's C-terminus can protect against Slack$^{G269S}$-induced seizures in mice. These insights about a Slack-Na$_V$1.6 complex challenge the traditional view of 'Slack as an isolated target' for anti-epileptic drug discovery efforts and can guide the development of innovative therapeutic strategies for KCNT1-related epilepsy.

## eLife assessment

The authors report that an interaction between the sodium-activated potassium channel Slack and Na$_V$1.6 sensitizes Slack to inhibition by quinidine. This is an **important** finding because it contributes to our understanding of how the antiseizure drug quinidine affects epilepsy syndromes arising from mutations in the Slack-encoding gene KCNT1. The results are largely **compelling** and the work will likely spark interest in further examining the proposed channel-channel interaction in neuronal cell membranes.

## Introduction

The sodium-activated potassium ($K_{Na}$) channels Slack and Slick were first identified in guinea pig cardiomyocytes and were subsequently found to be encoded by two genes of the *KCNT* (*SLO2*)

family (*Kameyama et al., 1984*; *Yuan et al., 2003*). Slack channels encoded by the *KCNT1* (*SLO2.2*) gene are gated by $Na^+$, while Slick channels encoded by the *KCNT2* (*SLO2.1*) gene are more sensitive to $Cl^-$ than $Na^+$. Slack channels are expressed at high levels in the central nervous system (CNS), especially the cortex and brainstem (*Joiner et al., 1998*; *Rizzi et al., 2016*; *Bhattacharjee et al., 2002*). Activation of Slack channels by intracellular sodium ions forms delayed outward currents in neurons, contributes to slow afterhyperpolarization (AHP) following repeated action potentials, and modulates the firing frequency of neurons (*Yang et al., 2007*; *Wallén et al., 2007*).

Mutations in the *KCNT1* gene have been implicated in a wide spectrum of epileptic disorders, including early-onset epilepsy (e.g., epilepsy of infancy with migrating focal seizures (EIMFS), non-EIMFS developmental and epileptic encephalopathies, and autosomal-dominant or sporadic sleep-related hypermotor epilepsy (ADSHE)) (*Heron et al., 2012*; *Barcia et al., 2012*; *Kingwell, 2012*; *Bonardi et al., 2021*). Over 50 mutations related to seizure disorders have been identified, typically displaying a gain-of-function (GOF) phenotype in heterologous expression systems (*Bonardi et al., 2021*; *McTague et al., 2018*). The prescribed antiarrhythmic drug, quinidine, has emerged as a precision therapy for KCNT1-related epilepsy by blocking Slack mutant variants in vitro and conferring decreased seizure frequency and improved psychomotor development in clinical treatment (*Milligan et al., 2014*; *Bearden et al., 2014*; *Mikati et al., 2015*; *Numis et al., 2018*). However, clinical quinidine therapy has shown limited success and contradictory therapeutic effects, probably due to poor blood–brain barrier penetration, dose-limiting off-target effects, phenotype–genotype associations, and rational therapeutic schedule (*Numis et al., 2018*; *Liu et al., 2023*; *Cole et al., 2021*; *Xu et al., 2022*).

Slack requires high intracellular-free $Na^+$ concentrations ($[Na^+]_{in}$) for its activation in neurons ($K_d$ of ~66 mM) (*Zhang et al., 2010*). However, previous investigations have shown that the $[Na^+]_{in}$ at resting states (~10 mM) is much lower than the $[Na^+]_{in}$ needed for effective Slack activation (e.g., $K_d$ value) (*Kameyama et al., 1984*; *Zhang et al., 2010*; *Budelli et al., 2009*). Therefore, native Slack channels need to localize with $Na^+$ sources within a nanodomain to be activated and exert physiological function. Slack channels are known to be functionally coupled with sodium-permeable ion channels in neurons, such as voltage-gated sodium ($Na_V$) channels and AMPA receptors (*Budelli et al., 2009*; *Hage and Salkoff, 2012*). A question arises as to whether these known $Na^+$ sources modulate Slack's sensitivity to quinidine blockade.

Here, we found that $Na_V1.6$ sensitizes Slack to quinidine blockade. Slack and $Na_V1.6$ form a complex that functions in $Na_V1.6$-mediated transient sodium influx to sensitize Slack to quinidine blockade in HEK293 cells and in primary cortical neurons. The widespread expression of these channel proteins in cerebral cortex, hippocampus, and cerebellum supports that the $Na_V1.6$-Slack complex is essential for the function of a wide range of electrically excitable neurons, and moreover, that this complex can be viewed as a vulnerable target for drug development to treat KCNT1-related disorders.

## Results

### $Na_V1.6$ sensitizes Slack to quinidine blockade

Slack currents are activated by sodium entry through voltage-gated sodium ($Na_V$) channels and ionotropic glutamate receptors (e.g., AMPA receptors) (*Budelli et al., 2009*; *Hage and Salkoff, 2012*; *Nanou et al., 2008*). To investigate potential modulators of Slack's sensitivity to quinidine blockade, we initially focused on the known $Na^+$ sources of Slack. Working in HEK293 cells, we co-expressed Slack with AMPA receptor subunits (GluA1, GluA2, GluA3, or GluA4) or $Na_V$ channel α subunits ($Na_V1.1$, $Na_V1.2$, $Na_V1.3$, or $Na_V1.6$), which are highly expressed in the CNS (*Goldin, 2001*; *Trimmer and Rhodes, 2004*; *Lai and Jan, 2006*; *Pickard et al., 2000*). The sensitivity of Slack to quinidine blockade was assessed based on the detected inhibitory effects of quinidine on delayed outward potassium currents (*Milligan et al., 2014*; *Cole et al., 2020*). Interestingly, all neuronal $Na_V$ channels significantly sensitized Slack to 30 µM quinidine blockade, whereas no effect was observed upon co-expression of Slack with GluA1, GluA2, GluA3, or GluA4 (*Figure 1—figure supplement 1*).

When co-expressing Slack with $Na_V1.6$ in HEK293 cells, $Na_V1.6$ sensitized Slack to quinidine blockade by nearly 100-fold ($IC_{50} = 85.13$ µM for Slack expressed alone and an $IC_{50} = 0.87$ µM for Slack upon co-expression with $Na_V1.6$) (*Figure 1A, B, D and F*). $Na_V1.6$ also exhibited greater than tenfold selectivity in sensitizing Slack to quinidine blockade against $Na_V1.1$, $Na_V1.2$, and $Na_V1.3$ (*Figure 1C*

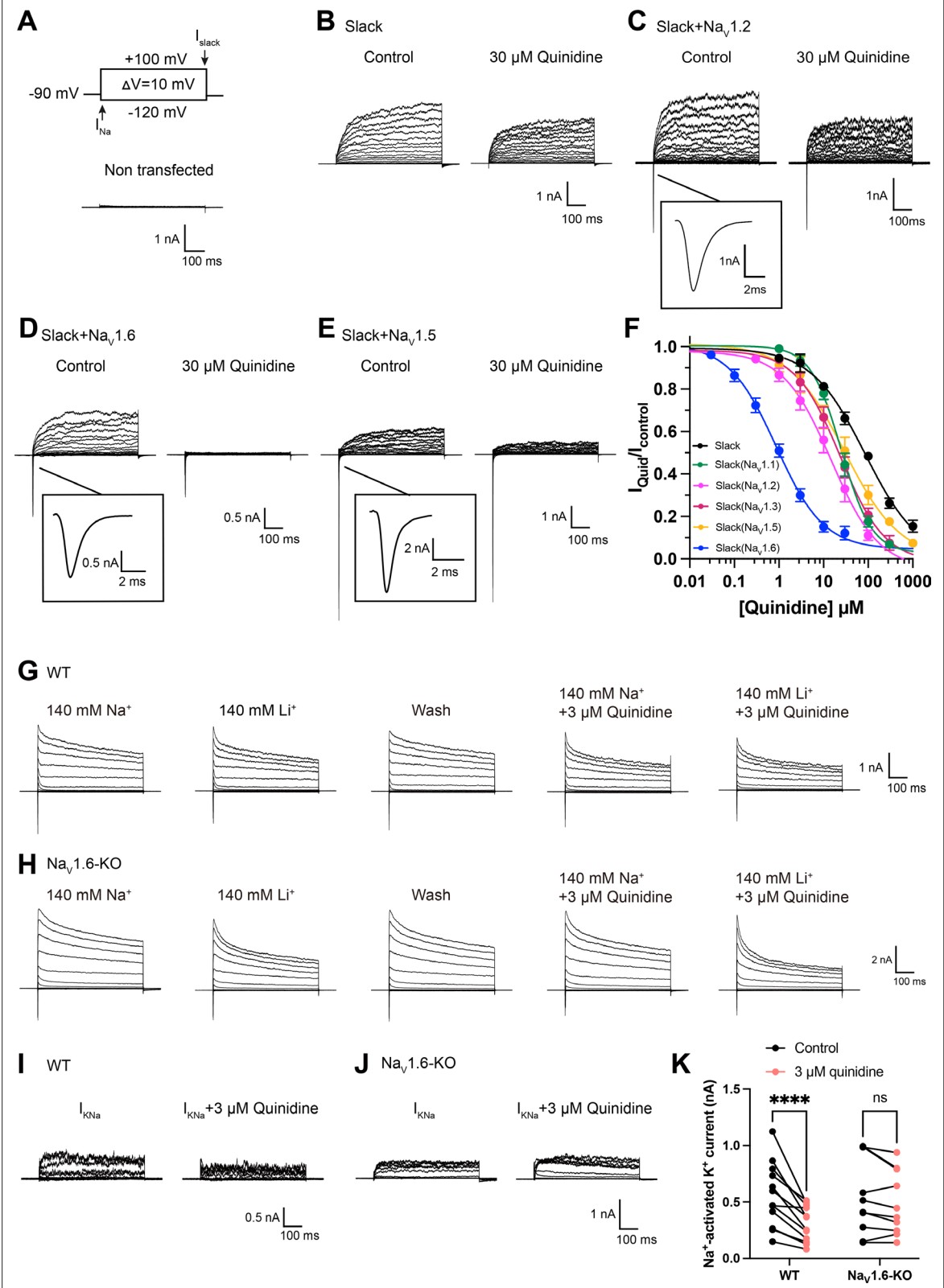

**Figure 1.** Na$_V$1.6 specifically sensitizes Slack to quinidine blockade. (**A**) The voltage protocol and current traces from control (non-transfected) HEK293 cells. The arrows on the voltage protocol indicate the onset of inward sodium currents through Na$_V$ channels and delayed outward potassium currents through Slack channels. The currents were evoked by applying 600 ms step pulses to voltages varying from –120 mV to +100 mV in 10 mV increments, with a holding potential of –90 mV and a stimulus frequency of 0.20 Hz. (**B**) Example current traces from HEK293 cells expressing Slack alone. The left

*Figure 1 continued on next page*

*Figure 1 continued*

traces show the family of control currents; the right traces show Slack currents remaining after application of 30 µM quinidine in the bath solution. (C–E) Example current traces from HEK293 cells co-expressing Slack with Na$_V$1.2 (C), Na$_V$1.6 (D), or Na$_V$1.5 (E) channels before and after application of 30 µM quinidine. (F) The concentration–response curves for blocking of Slack by quinidine at +100 mV upon expression of Slack alone (n = 6) and co-expression of Slack with Na$_V$1.1 (n = 7), Na$_V$1.2 (n = 10), Na$_V$1.3 (n = 13), Na$_V$1.5 (n = 9), or Na$_V$1.6 (n = 19). Please refer to *Supplementary file 1a* for IC$_{50}$ values. (G, H) Delayed outward currents in primary cortical neurons from postnatal (P0-P1) homozygous Na$_V$1.6 knockout C3HeB/FeJ mice (Na$_V$1.6-KO) (H) and the wild-type littermate controls (WT) (G). Current traces were elicited by 600 ms step pulses to voltages varying from –120 mV to +100 mV in 20 mV increments, with a holding potential of –70 mV, and recorded with different bath solutions in the following order: Na$^+$-based bath solution (I$_{Control}$), replacement of external Na$^+$ with Li$^+$ in equivalent concentration (I$_{Li}$), washout of quinidine by Na$^+$-based bath solution (I$_{Wash}$), Na$^+$-based bath solution with 3 µM quinidine (I$_{Quid}$), Li$^+$-based bath solution with 3 µM quinidine (I$_{Li+Quid}$). The removal and subsequent replacement of extracellular Na$^+$ revealed the I$_{KNa}$ in neurons. (I, J) The sensitivity of native sodium-activated potassium currents (I$_{KNa}$) to 3 µM quinidine blockade in WT (I) and Na$_V$1.6-KO (J) neurons. I$_{KNa}$ before application of quinidine was obtained from the subtraction of I$_{Control}$ and I$_{Li}$. Maintained I$_{KNa}$ after application of 3 µM quinidine was obtained from the subtraction of I$_{Quid}$ and I$_{Li+Quid}$. (K) Summarized amplitudes of I$_{KNa}$ before and after application of 3 µM quinidine in the bath solution in WT (black, n = 12) and Na$_V$1.6-KO (red, n = 10) primary cortical neurons. ****p<0.0001, two-way repeated-measures ANOVA followed by Bonferroni's post hoc test.

The online version of this article includes the following source data and figure supplement(s) for figure 1:

**Source data 1.** Numerical data for *Figure 1* and *Figure 1—figure supplements 1–3*.

**Figure supplement 1.** The sensitivity of Slack to quinidine blockade upon expression of Slack alone and co-expression of Slack with sodium-permeable channels.

**Figure supplement 2.** The sensitivity of Slick to quinidine blockade upon expression of Slick alone or co-expression of Slick with Na$_V$1.6.

**Figure supplement 3.** The amplitudes and sensitivity to quinidine of sodium-activated potassium currents in primary cortical neurons.

*and F*, *Supplementary file 1a*). When we co-expressed the cardiac sodium channel Na$_V$1.5 with Slack, we observed only an approximately threefold sensitization of Slack to quinidine blockade (*Figure 1E and F*). These results together indicate that the apparent functional coupling between Slack and Na$_V$ channels is Na$_V$-channel-subtype-specific, with Na$_V$1.6 being particularly impactful in Slack's responsivity to quinidine blockade.

Slack and Slick are both K$_{Na}$ channels (with 74% sequence identity) and adopt similar structures (*Kaczmarek, 2013*), and Slick is also blocked by quinidine (*Bhattacharjee et al., 2003*). We next assessed whether Na$_V$1.6 sensitizes Slick to quinidine blockade and observed that, similar to Slack, co-expression of Slick and Na$_V$1.6 in HEK293 cells resulted in a sensitization of Slick to quinidine blockade (sevenfold) (*Figure 1—figure supplement 2A and B*). These results support that Na$_V$1.6 regulates both Slack and Slick and that Na$_V$1.6 can sensitize K$_{Na}$ channels to quinidine blockade in vitro.

We also asked whether Na$_V$1.6 sensitizes native K$_{Na}$ channels to quinidine blockade in neurons. We performed whole-cell patch-clamp recordings in primary cortical neurons from postnatal (P0-P1) homozygous Na$_V$1.6 knockout C3HeB/FeJ mice and the wild-type littermate controls. As lithium is a much weaker activator of K$_{Na}$ channels than sodium (*Kaczmarek, 2013*), K$_{Na}$ currents (I$_{KNa}$) were isolated by replacing sodium ions with equivalent lithium ions in the bath solution (*Figure 1G and H*) as previously described (*Hage and Salkoff, 2012*). The amplitudes of I$_{KNa}$ remained unaffected by the homozygous knockout of Na$_V$1.6 (*Figure 1—figure supplement 3A*). Notably, 3 µM quinidine significantly inhibited native I$_{KNa}$ (44%) in wild-type neurons (*Figure 1I and K*) and this inhibitory effect was not limited to targeting the larger I$_{KNa}$ (*Figure 1—figure supplement 3B*). Conversely, the same concentration of quinidine had no significant effect on I$_{KNa}$ in Na$_V$1.6-knockout (Na$_V$1.6-KO) neurons (*Figure 1J and K*). These results support that Na$_V$1.6 is required for the observed high sensitivity of native K$_{Na}$ channels to quinidine blockade.

## Transient sodium influx through Na$_V$1.6 enhances Na$_V$1.6-mediated sensitization of Slack to quinidine blockade

We next investigated the biomolecular mechanism underlying Na$_V$1.6-mediated sensitization of Slack to quinidine blockade. Considering that Slack currents are activated by sodium influx (*Hage and Salkoff, 2012*; *Kaczmarek, 2013*), we initially assessed the effects of Na$_V$1.6-mediated sodium influx on sensitizing Slack to quinidine blockade. We used 100 nM tetrodotoxin (TTX) to block Na$_V$1.6-mediated sodium influx (*Figure 2—figure supplement 1A and B*; *Rosker et al., 2007*). In HEK293 cells expressing Slack alone, 100 nM TTX did not affect Slack currents; nor did it affect Slack's sensitivity

to quinidine blockade ($IC_{50}$ = 83.27 µM) (*Figure 2A and B*, *Figure 2—figure supplement 1C and D*). In contrast, upon co-expression of Slack and $Na_V1.6$ in HEK293 cells, bath application of 100 nM TTX significantly reduced the effects of $Na_V1.6$ in sensitizing Slack to quinidine blockade ($IC_{50}$ = 25.04 µM) (*Figure 2A and B*). These findings support that sodium influx through $Na_V1.6$ contributes to $Na_V1.6$-mediated sensitization of Slack to quinidine blockade.

It is known that $Na_V1.6$-mediated sodium influx involves a transient inward flux that reaches a peak before subsequently decaying to the baseline within a few milliseconds; this is termed a transient sodium current ($I_{NaT}$) (*Rush et al., 2005*). A small fraction of $Na_V1.6$ currents are known to persist after the rapid decay of $I_{NaT}$, and these are termed persistent sodium currents ($I_{NaP}$) (*Chatelier et al., 2010*). We isolated $I_{NaT}$ and $I_{NaP}$ to explore their potential contributions in sensitizing Slack to quinidine blockade. We selectively inactivated $I_{NaT}$ using a depolarized prepulse of –40 mV (*Figure 2—figure supplement 2A*) and selectively blocked $I_{NaP}$ by bath application of 20 µM riluzole, which is a relatively specific $I_{NaP}$ blocker that is known to stabilize inactivated-state $Na_V$ channels and delay recovery from inactivation (*Urbani and Belluzzi, 2000*; *Lukacs et al., 2018*). Our findings ultimately confirmed that the 20 µM riluzole selectively blocked $I_{NaP}$ compared to $I_{NaT}$ in HEK293 cells co-expressing Slack and $Na_V1.6$ (*Figure 2—figure supplement 2B*) and that 20 µM riluzole had no effect on Slack currents when expressed alone (*Figure 2—figure supplement 2C and D*).

Consistent with previous investigations (*Budelli et al., 2009*; *Hage and Salkoff, 2012*), inactivating $I_{NaT}$ reduced whole-cell Slack currents by 20%, and blocking $I_{NaP}$ reduced Slack currents by 40% at +100 mV (*Figure 2C, D, F and G*), supporting that $Na_V1.6$-mediated sodium influx activates Slack. Interestingly, inactivating $I_{NaT}$ resulted in a >20-fold decrease in Slack's sensitization to quinidine blockade ($IC_{50}$ = 22.26 µM) (*Figure 2C and E*). In contrast, blocking $I_{NaP}$ had no effect on Slack's sensitization to quinidine blockade ($IC_{50}$ = 1.60 µM) (*Figure 2F and H*). These findings indicate that $Na_V1.6$ sensitizes Slack to quinidine blockade via $I_{NaT}$ but not $I_{NaP}$.

Given that Slack current amplitudes are sensitive to sodium influx, and considering that quinidine is a sodium channel blocker, we examined whether $Na_V1.6$ has higher sensitivity to quinidine blockade than other $Na_V$ channel subtypes, which could plausibly explain the observed increased strength of sensitization. We used whole-cell patch-clamping to assess the sensitivity of $Na_V1.1$, $Na_V1.2$, $Na_V1.3$, $Na_V1.5$, and $Na_V1.6$ to quinidine blockade. These sodium channels exhibited similar levels of quinidine sensitivity ($IC_{50}$ values in the range of 35.61–129.84 µM) (*Figure 2I and J* and *Supplementary file 1b*), all of which were at least 40-fold lower than the $Na_V1.6$-mediated sensitization of Slack to quinidine blockade (*Figure 1F*). Additionally, co-expressing Slack with $Na_V1.1$, $Na_V1.2$, $Na_V1.3$, $Na_V1.5$, or $Na_V1.6$ in HEK293 cells did not change the sensitivity of these $Na_V$ channel subtypes to quinidine blockade (*Figure 2—figure supplement 3* and *Supplementary file 1b*). Thus, differential quinidine affinity for specific $Na_V$ channel subtypes cannot explain the large observed $Na_V1.6$-mediated sensitization of Slack to quinidine blockade. Moreover, it is clear that $Na_V1.6$-mediated sensitization of Slack to quinidine blockade is directly mediated by $I_{NaT}$, rather than through some secondary effects related to $Na_V1.6$'s higher sensitivity to quinidine blockade.

## Slack physically interacts with $Na_V1.6$

We found that the specific voltage-gated sodium channel blocker TTX did not completely abolish the effects of $Na_V1.6$ on sensitizing Slack to quinidine blockade (*Figure 2B*), so it appears that a sodium-influx-independent mechanism is involved in the observed $Na_V1.6$-mediated sensitization of Slack to quinidine blockade. We therefore investigated a potential physical interaction between Slack and $Na_V1.6$. We initially assessed the cellular distribution of $Na_V1.2$, $Na_V1.6$, and Slack in the hippocampus and the neocortex of mouse. Consistent with previous studies (*Hu et al., 2009*; *Liu et al., 2017*), $Na_V1.2$ and $Na_V1.6$ were localized to the axonal initial segment (AIS) of neurons, evident as the co-localization of $Na_V$ and AnkG, a sodium channel-associated protein known to accumulate at the AIS (*Figure 3A*). Slack channels were also localized to the AIS of these neurons (*Figure 3A*), indicating that Slack channels are located in close proximity to $Na_V1.6$ channels, and supporting their possible interaction. Moreover, $Na_V1.6$ was co-immunoprecipitated with Slack in homogenates from mouse cortical and hippocampal tissues and from HEK293T cells co-transfected with Slack and $Na_V1.6$ (*Figure 3B and C*), supporting that Slack and $Na_V1.6$ form protein complexes in mouse brains.

To assess the interaction between Slack and $Na_V1.6$ inside living cells, we performed a FRET assay in transfected HEK293T cells (*Takanishi et al., 2006*). Briefly, we genetically fused mTFP1 and mVenus

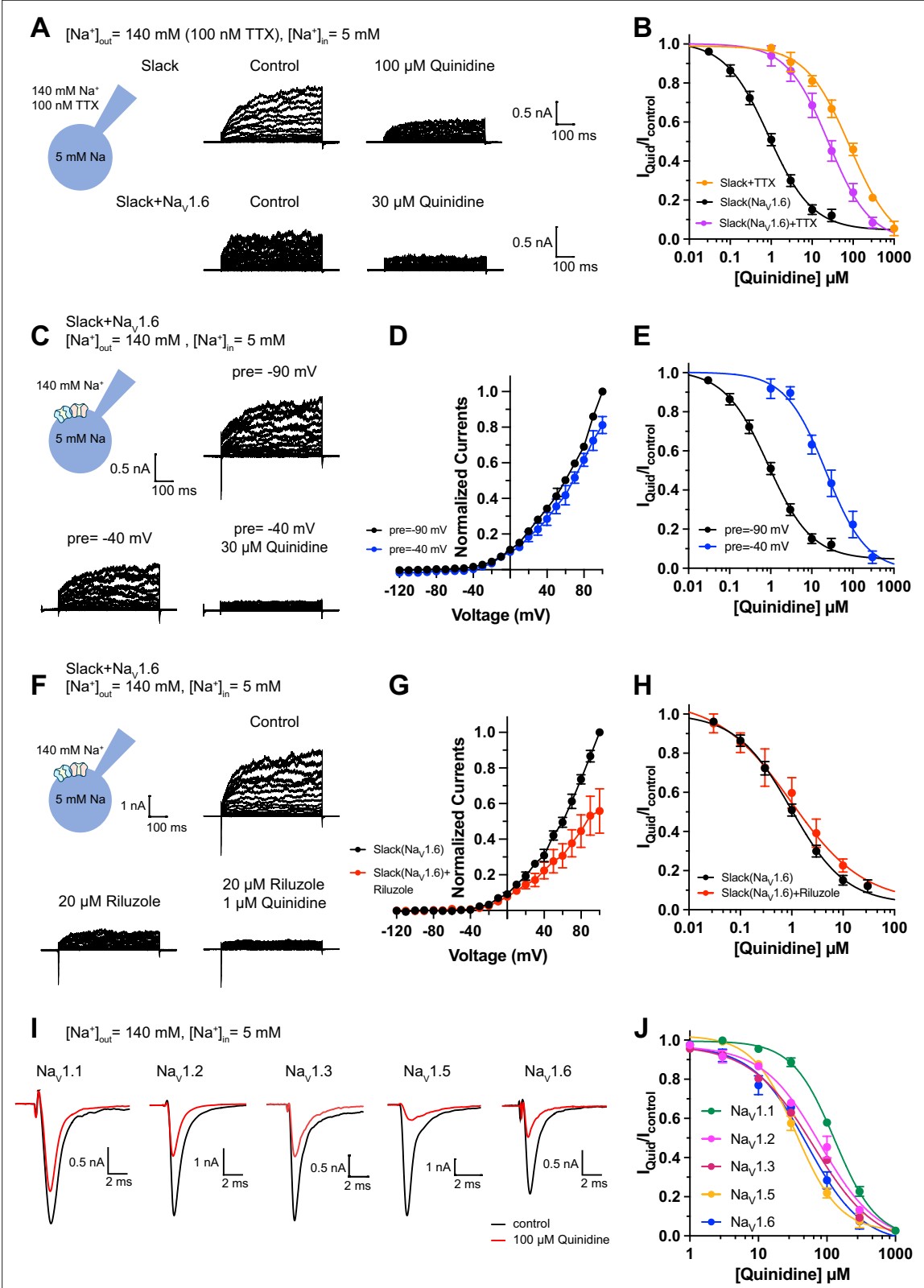

**Figure 2.** Blocking transient sodium influx through Na$_V$1.6 reduces Na$_V$1.6-mediated sensitization of Slack to quinidine blockade. (**A**) Example current traces from HEK293 cells expressing Slack alone (top) and co-expressing Slack with Na$_V$1.6 (bottom), with 100 nM tetrodotoxin (TTX) in the bath solution. The left traces show the family of control currents; the right traces show Slack currents remaining after application of quinidine. The presented concentrations of quinidine were chosen to be near the IC$_{50}$ values. (**B**) The concentration–response curves for blocking of Slack by quinidine at +100 mV

*Figure 2 continued on next page*

*Figure 2 continued*

upon expression of Slack alone (n = 3) and co-expression of Slack with $Na_V1.6$ (n = 7), with 100 nM TTX in the bath solution. (**C**) Top: example current traces recorded from a HEK293 cell co-expressing Slack with $Na_V1.6$ and evoked from a 100 ms prepulse (pre) of –90 mV, with the same voltage protocol as in **Figure 1D**. Bottom: example current traces recorded from the same cell but evoked from a 100 ms prepulse of –40 mV, before and after application of quinidine. (**D**) I–V curves of Slack upon co-expression with $Na_V1.6$. The currents were evoked from a prepulse of –90 mV (black) or –40 mV (blue). (**E**) The concentration–response curves for blocking of Slack by quinidine at +100 mV with a prepulse of –90 mV (black, n = 19) or –40 mV (blue, n = 5). (**F**) Top: example current traces recorded from a HEK293 cell co-expressing Slack and $Na_V1.6$ without riluzole in the bath solution. Bottom: example current traces recorded from the same cell with 20 μM riluzole in the bath solution, before and after application of quinidine. (**G**) I–V curves of Slack upon co-expression with $Na_V1.6$ before (black) and after (red) application of 20 μM riluzole into bath solution. The concentration–response curves for blocking of Slack by quinidine upon co-expression of Slack with $Na_V1.6$, without (n = 19) or with (n = 6) 20 μM riluzole in the bath solution. (**I, J**) The sensitivity of $Na_V$ channel subtypes to quinidine blockade upon expression of $Na_V$ alone in HEK293 cells. Example current traces (**I**) were evoked by a 50 ms step depolarization to 0 mV from a holding potential of –90 mV. The concentration–response curves for blocking of $Na_V$ channel subtypes by quinidine (**J**) are shown on the right panel (n = 5 for $Na_V1.1$, n = 3 for $Na_V1.2$, n = 6 for $Na_V1.3$, n = 6 for $Na_V1.5$, and n = 4 for $Na_V1.6$).

The online version of this article includes the following source data and figure supplement(s) for figure 2:

**Source data 1.** Numerical data for *Figure 2* and *Figure 2—figure supplements 1–4*.

**Figure supplement 1.** Effects of 100 nM tetrodotoxin (TTX) on $Na_V1.6$ and Slack currents.

**Figure supplement 2.** Effects of depolarized prepulse potentials and riluzole on channels.

**Figure supplement 3.** The sensitivity of $Na_V$ channel subtypes to quinidine blockade.

**Figure supplement 4.** The dynamic properties of Slack and $Na_V1.6$ channels.

to the C-terminal regions of Slack and $Na_V1.6$, respectively (**Figure 3D**). Upon imaging the emission spectra cells co-expressing $Na_V1.6$-mVenus and Slack-mTFP1 (measured at the plasma membrane region) (**Figure 3E**), we detected positive FRET signals, indicating a Slack-$Na_V1.6$ interaction (**Figure 3F and H**). The plasma membrane regions from HEK293T cells co-transfected with $Na_V1.6$ and Slack showed FRET efficiency values much larger than a negative control (in which standalone mVenus and mTFP1 proteins were co-expressed) (**Figure 3G and H**), indicating that Slack channels reside in close spatial proximity (less than 10 nm) to $Na_V1.6$ channels in living cells.

We next characterized the consequences of the Slack-$Na_V1.6$ interaction in HEK293 cells using whole-cell recordings. Slack increased the rate of recovery from fast inactivation of $Na_V1.6$ (**Figure 2—figure supplement 4E**), with no significant effects on the steady-state activation, steady-state fast inactivation, or ramp currents (**Figure 2—figure supplement 4C, D and F** and **Supplementary file 1c**). Additionally, we found that $Na_V1.6$ had no significant effects on the activation rate or the current–voltage (I–V) relationship of Slack currents (**Figure 2—figure supplement 4A and B**). These results indicate that the physical interaction between Slack and $Na_V1.6$ produces functional consequences. Taken together, these findings support functional and physical coupling of Slack and $Na_V1.6$.

## $Na_V1.6$'s N- and C-termini bind to Slack's C-terminus and sensitize Slack to quinidine blockade

To explore whether the physical interaction between Slack and $Na_V1.6$ is required for sodium-influx-mediated sensitization of Slack to quinidine blockade, we performed inside-out patch-clamp recordings on HEK293 cells transfected with Slack alone or co-transfected with Slack and $Na_V1.6$. Note that in these experiments the intracellular sodium concentration ($[Na^+]_{in}$) was raised to 140 mM (a concentration at which most Slack channels can be activated; *Zhang et al., 2010*), seeking to mimic the increased intracellular sodium concentration upon sodium influx. When expressing Slack alone, increasing the sodium concentration did not sensitize Slack to quinidine blockade ($IC_{50}$ = 120.42 μM) (**Figure 4A**). However, upon co-expression of $Na_V1.6$ and Slack, $Na_V1.6$ significantly sensitized Slack to quinidine blockade ($IC_{50}$ = 2.91 μM) (**Figure 4A**). These results support that physical interaction between Slack and $Na_V1.6$ is a prerequisite for sodium-influx-mediated sensitization of Slack to quinidine blockade.

To investigate which interacting domains mediate $Na_V1.6$'s sensitization of Slack to quinidine blockade, we focused on $Na_V1.6$'s cytoplasmic fragments, including its N-terminus, inter-domain linkers, and C-terminus (**Figure 4B**). Whole-cell recordings from HEK293 cells co-expressing Slack with these $Na_V1.6$ fragments revealed that the N-terminus and C-terminus of $Na_V1.6$ significantly enhanced the sensitivity of Slack to quinidine blockade ($IC_{50}$ = 31.59 μM for Slack upon co-expression with

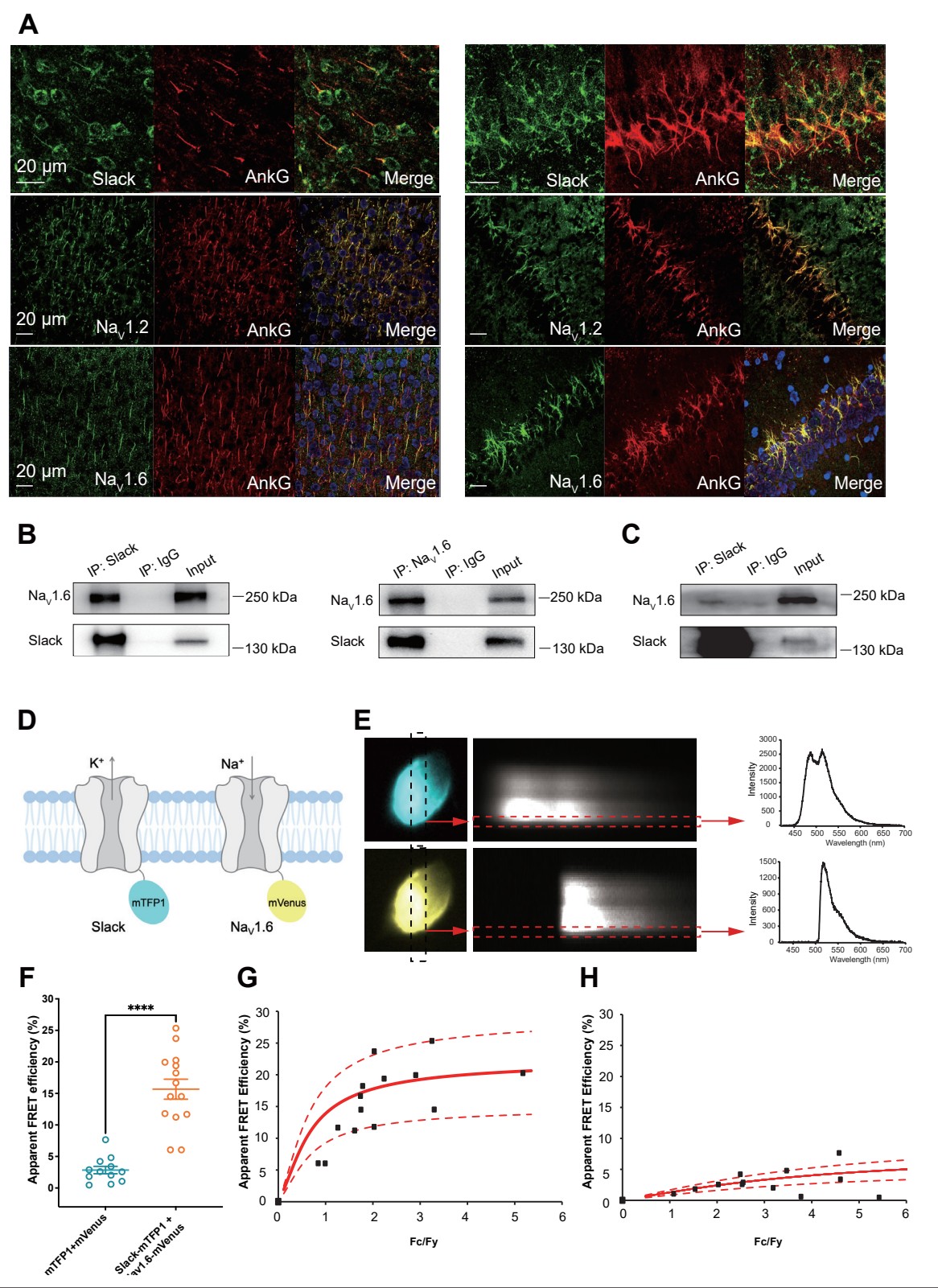

**Figure 3.** Slack physically interacts with Na$_V$1.6. (**A**) Immunofluorescence of Slack, Na$_V$1.2, Na$_V$1.6 (green), and AnkG (red) in neocortex layer 5 (left) and hippocampal CA1 pyramidal cell layer (right). Confocal microscopy images were obtained from coronal brain slices of C57BL/6 mice. The panels from top to bottom show the double staining of Slack with AnkG, Na$_V$1.2 with AnkG, and Na$_V$1.6 with AnkG, respectively. DAPI (blue) was used for nuclear counterstaining. (**B**) Coimmunoprecipitation (Co-IP) of Slack and Na$_V$1.6 in cell lysates from HEK293T cells co-transfected with Slack and Na$_V$1.6. (**C**) Co-IP

*Figure 3 continued on next page*

*Figure 3 continued*

of Slack and $Na_V1.6$ in mouse brain tissue lysates. Input volume corresponds to 10% of the total lysates for Co-IP. (**D**) A schematic diagram showing the fluorescence-labeled Slack and $Na_V1.6$. mTFP1 and mVenus were fused to the C-terminal region of Slack (Slack-mTFP1) and $Na_V1.6$ ($Na_V1.6$-mVenus), respectively. (**E**) FRET imaging of Slack-mTFP1 and $Na_V1.6$-mVenus co-expressed in HEK293 cells. The emission spectra measured from the edge of cell (dotted arrows in red) are used for FRET efficiency calculation. (**F**) The apparent FRET efficiency measured from cells co-expressing the fluorophore-tagged ion channels (n=14) or co-expressing the fluorophores (n=12). ****$p<0.0001$, Mann–Whitney test. (**G, H**) The FRET efficiency measured from cells co-expressing the fluorophore-tagged ion channels (**G**), or from cells co-expressing fluorophores (**H**). The efficiency value was plotted as a function of the fluorescence intensity ratio between mTFP1 and mVenus (Fc/Fy). Each symbol represents a single cell. The solid curve represents the FRET model that yields the best fit; dotted curves represent models with 5% higher or lower FRET efficiencies.

The online version of this article includes the following source data and figure supplement(s) for figure 3:

**Source data 1.** Numerical data for *Figure 3*.

**Figure supplement 1.** Glutathione S-transferase (GST) pull-down assay of Slack with the N- and C-termini of $Na_V1.6$.

$Na_V1.6$'s N-terminus and $IC_{50}$ = 43.70 μM for Slack upon co-expression with $Na_V1.6$'s C-terminus); note that the inter-domain linkers had no effect (*Figure 4C*).

Subsequent co-immunoprecipitation and glutathione S-transferase (GST) pull-down assays of HEK293T cell lysates experimentally confirmed that $Na_V1.6$'s N- and C-termini each interact with Slack (*Figure 4D* and *Figure 3—figure supplement 1*). Additionally, whole-cell recordings using an $[Na^+]_{in}$ of 5 mM again showed that $Na_V1.6$'s N-terminus and $Na_V1.6$'s C-terminus sensitize Slack to quinidine blockade ($IC_{50}$ = 27.87 μM) (*Figure 4E*). And inside-out recordings using an $[Na^+]_{in}$ of 140 mM showed that co-expression of Slack, $Na_V1.6$'s N-terminus, and $Na_V1.6$'s C-terminus resulted in obvious sensitization of Slack to quinidine blockade, with an $IC_{50}$ of 2.57 μM (*Figure 4F*), thus fully mimicking the aforementioned effects of full-length $Na_V1.6$ ($IC_{50}$ = 2.91 μM) (*Figure 4A*). These findings support that the binding of $Na_V1.6$'s N- and C-termini to Slack is required for $Na_V1.6$'s sensitization of Slack to quinidine blockade.

Recalling that $Na_V1.5$ had the least pronounced effect in sensitizing Slack to quinidine blockade among all examined $Na_V$ channels ($IC_{50}$ = 29.46 μM) (*Figure 1F* and *Supplementary file 1a*), we initially compared the sodium currents of $Na_V$ channel subtypes co-expressed with Slack. The results revealed no significant differences in activation time constants (*Figure 4—figure supplement 1A*). It is worth noting that $Na_V1.5$ exhibited significantly larger current amplitudes than $Na_V1.6$ (*Figure 4—figure supplement 1B*). Subsequently, we constructed $Na_V1.5$–1.6 chimeras to test the roles of $Na_V1.6$'s N- and C-termini in sensitizing Slack to quinidine blockade. The replacement of both the N-terminus (residues 1–131) and C-terminus (residues 1772–2016) of $Na_V1.5$ with $Na_V1.6$'s N-terminus (residues 1–132) and C-terminus (residues 1766–1980) (namely $Na_V1.5/6^{NC}$) fully mimicked effects of $Na_V1.6$ in sensitizing Slack to quinidine blockade ($IC_{50}$ = 1.13 μM) (*Figure 4G–I*). We also found that replacement of $Na_V1.5$'s N-terminus (residues 1–131) with $Na_V1.6$'s N-terminus (residues 1–132) (namely $Na_V1.5/6^N$) fully mimicked the effects of $Na_V1.6$ ($IC_{50}$ = 1.18 μM) (*Figure 4G–I*).

Notably, despite both $Na_V1.5$ and $Na_V1.5/6^N$ exhibiting much larger current amplitudes compared to $Na_V1.6$ (*Figure 4—figure supplement 1B*), only $Na_V1.5/6^N$ replicated the effect of $Na_V1.6$ in sensitizing Slack to quinidine blockade (*Figure 4H and I*). These results suggest that the observed differences between $Na_V1.5$ and $Na_V1.6$ in sensitizing Slack are unlikely to be attributed to $Na_V1.6$'s smaller sodium current amplitudes but involve $Na_V1.6$'s N-terminus. Consistently, $Na_V1.5$'s C-terminus sensitized Slack to quinidine blockade ($IC_{50}$ = 37.59 μM), whereas $Na_V1.5$'s N-terminus had no effect on Slack sensitization (*Figure 4C*). These findings support that $Na_V1.6$'s N-terminus is essential for sensitizing Slack to quinidine blockade.

Having demonstrated that $Na_V1.6$ sensitizes Slack via $Na_V1.6$'s cytoplasmic N- and C-termini, we investigated which domains of Slack interact with $Na_V1.6$ and focused on Slack's cytoplasmic fragments, including Slack's N-terminus and C-terminus (*Figure 5A*). In HEK293 cells co-expressing $Na_V$ and Slack, $Na_V$-mediated sensitization of Slack to quinidine blockade was significantly attenuated upon the additional expression of Slack's C-terminus, but not of Slack's N-terminus (*Figure 5B and C*), suggesting that Slack's C-terminus can disrupt the Slack-$Na_V1.6$ interaction by competing with Slack for binding to $Na_V1.6$. Consistently, Slack's C-terminus co-immunoprecipitated with $Na_V1.6$'s N- and C-termini in HEK293T cell lysates (*Figure 5D*). Together, these results support that Slack's C-terminus physically interacts with $Na_V1.6$ and that this interaction is required for $Na_V1.6$'s sensitization of Slack to quinidine blockade.

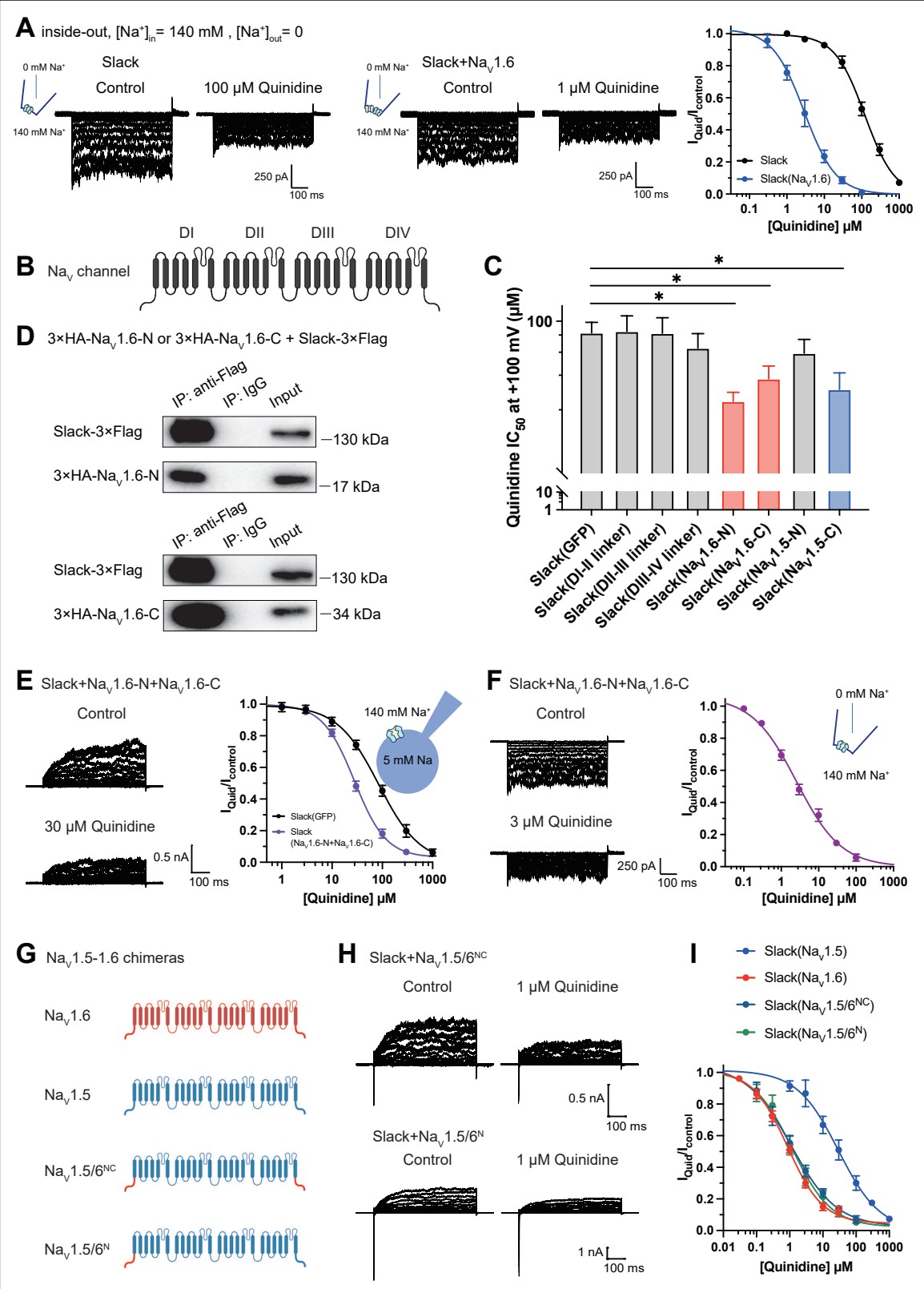

**Figure 4.** $Na_V1.6$'s N- and C-termini interacting with Slack is a prerequisite for $Na_V1.6$-mediated sensitization of Slack to quinidine blockade. (**A**) The sensitivity of Slack to quinidine blockade upon expression of Slack alone (n = 3) and co-expression of Slack with $Na_V1.6$ (n = 3) from excised inside-out patches. The pipette solution contained (in mM) 130 KCl, 1 EDTA, 10 HEPES, and 2 $MgCl_2$ (pH 7.3); the bath solution contained (in mM) 140 NaCl, 1 EDTA, 10 HEPES, and 2 $MgCl_2$ (pH 7.4). The membrane voltage was held at 0 mV and stepped to voltages varying from −100 mV to 0 mV in 10 mV

*Figure 4 continued on next page*

*Figure 4 continued*

increments. Example current traces are shown on the left panel. The concentration–response curves for blocking of Slack by quinidine are shown on the right panel. (**B**) Domain architecture of the human $Na_V$ channel pore-forming α subunit. (**C**) Calculated $IC_{50}$ values at +100 mV of quinidine on Slack upon co-expression with indicate cytoplasmic fragments from $Na_V$ channels. For $Na_V1.6$, cytoplasmic fragments used include N-terminus ($Na_V1.6$-N, residues 1–132), inter-domain linkers (domain I–II linker, residues 409–753; domain II–III linker, residues 977–1199; domain III–IV linker, residues 1461–1523), and C-terminus ($Na_V1.6$-C, residues 1766–1980). For $Na_V1.5$, cytoplasmic fragments used include N-terminus ($Na_V1.5$-N, residues 1–131) and C-terminus ($Na_V1.5$-C, residues 1772–2016). (**D**) Co-IP of Slack and terminal domains of $Na_V1.6$ in cell lysates from HEK293T cells co-expressing 3×Flag-tagged Slack (Slack-3×Flag) and 3×HA-tagged termini of $Na_V1.6$ (3×HA-$Na_V1.6$-N or 3×HA-$Na_V1.6$-C). The 3×Flag tag was fused to the C-terminal region of Slack and the 3×HA tag was fused to the N-terminal region of $Na_V1.6$'s fragments. (**E**) The sensitivity of Slack to quinidine blockade upon co-expression of Slack with GFP (n = 12) or N- and C-termini of $Na_V1.6$ (n = 11), from whole-cell recordings. (**F**) The sensitivity of Slack to quinidine blockade upon co-expression of Slack with N- and C-termini of $Na_V1.6$, from excised inside-out recordings (n = 10, using the same protocols as in A). Example current traces before and after application of quinidine are shown on the left panel. The concentration–response curves are shown on the right panel. (**G**) A schematic diagram of the $Na_V1.5$–1.6 chimeric channels ($Na_V1.5/6^{NC}$ and $Na_V1.5/6^{N}$) used in this study. (**H**) Example current traces recorded from HEK293 cells co-expressing Slack and $Na_V1.5$–1.6 chimeras before and after application of the indicated concentration of quinidine. (**I**) The concentration–response curves for blocking of Slack by quinidine upon co-expression of Slack with $Na_V1.5$ (n = 9), $Na_V1.6$ (n = 19), $Na_V1.5/6^{NC}$ (n = 9), or $Na_V1.5/6^{N}$ (n = 9).

The online version of this article includes the following source data and figure supplement(s) for figure 4:

**Source data 1.** Numerical data for *Figure 4* and *Figure 4—figure supplement 1*.

**Figure supplement 1.** Comparison of the activation and amplitudes of $Na_V$ channel subtypes currents upon co-expressed with Slack.

## $Na_V1.6$ binds to and sensitizes epilepsy-related Slack mutant variants to quinidine blockade

Over 50 mutations in KCNT1 (Slack) have been identified and related to seizure disorders (*Bonardi et al., 2021*). Having established that $Na_V1.6$ can sensitize wild-type Slack to quinidine blockade, we next investigated whether $Na_V1.6$ also sensitizes epilepsy-related Slack mutant variants to quinidine blockade. We chose three Slack pathogenic mutant variants (K629N, R950Q, and K985N) initially detected in patients with KCNT1-related epilepsy (*Mikati et al., 2015*; *Dilena et al., 2018*; *Abdel-nour et al., 2018*). Considering that these three mutations are located in Slack's C-terminus, and recalling that Slack's C-terminus interacts with $Na_V1.6$ (*Figure 5D*), we first used co-immunoprecipitation assays and successfully confirmed that each of these Slack mutant variants interacts with $Na_V1.6$ in HEK293T cell lysates (*Figure 6A*).

Subsequently, whole-cell recordings revealed that the examined Slack mutant variants exhibited no discernible impact on the amplitudes of $Na_V1.6$ currents (*Figure 6—figure supplement 1A*), while pharmacologically blocking $Na_V1.6$ currents using bath application of 100 nM TTX significantly reduced the amplitudes of Slack mutant variant ($Slack^{R950Q}$) currents (*Figure 6—figure supplement 1B*), suggesting a $Na^+$-mediated functional coupling between Slack mutant variant and $Na_V1.6$ currents. Notably, $Na_V1.6$ significantly sensitized all of the examined Slack mutant variants to quinidine blockade, with $IC_{50}$ values ranging from 0.26 to 2.41 µM (*Figure 6B–D* and *Supplementary file 1d*). These results support that $Na_V1.6$ interacts with examined Slack mutant variants and sensitizes them to quinidine blockade. It is plausible that the Slack-$Na_V1.6$ interaction contributes to the therapeutical role of quinidine in the treatment of KCNT1-related epilepsy.

## Viral expression of Slack's C-terminus prevents $Slack^{G269S}$-induced seizures

Having established that blocking $Na_V1.6$-mediated sodium influx significantly reduced Slack current amplitudes (*Figure 2D and G* and *Figure 6—figure supplement 1B*), we found that the heterozygous knockout of $Na_V1.6$ significantly reduced the AHP amplitude in murine hippocampal neurons (*Figure 7—figure supplement 1*), together indicating that $Na_V1.6$ activates native Slack through providing $Na^+$. We therefore assumed that disruption of the Slack-$Na_V1.6$ interaction should reduce the amount of $Na^+$ in the close vicinity of epilepsy-related Slack mutant variants and thereby counter the increased current amplitudes of these Slack mutant variants. Pursuing this, we disrupted the Slack-$Na_V$ interaction by overexpressing Slack's C-terminus (to compete with Slack) and measured whole-cell current densities. In HEK293 cells co-expressing epilepsy-related Slack mutant variants (G288S, R398Q) (*Rizzo et al., 2016*; *Barcia et al., 2019*) and $Na_V1.5/6^{NC}$, expression of Slack's C-terminus significantly reduced whole-cell current densities of $Slack^{G269S}$ and $Slack^{R398Q}$ (*Figure 7A and B*),

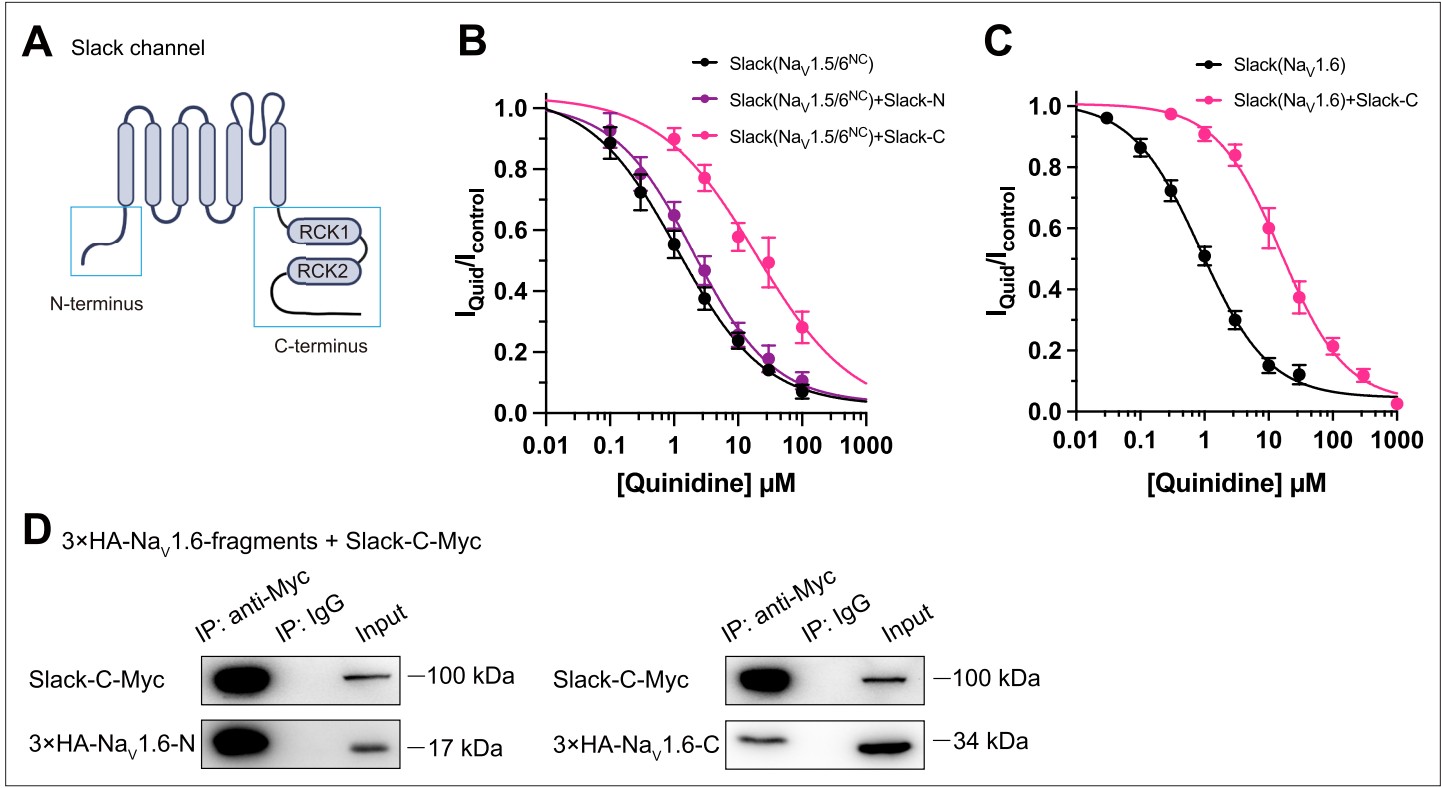

**Figure 5.** Slack's C-terminus is required for Na$_V$1.6-mediated sensitization of Slack to quinidine blockade. (**A**) Domain architecture of the human Slack channel subunit. Slack's N-terminus (Slack-N, residues 1–116) and C-terminus (Slack-C, residues 345–1235) are shown in the blue boxes. (**B**) The concentration–response curves for blocking of Slack by quinidine upon additional expression of Slack's N- or C-terminus in HEK293T cells co-expressing Slack and Na$_V$1.5/6$^{NC}$. (**C**) The concentration–response curves for blocking Slack by quinidine upon additional expression of Slack's C-terminus in HEK293 cells co-expressing Slack and Na$_V$1.6. (**D**) Co-IP of Myc-tagged Slack's C-terminus (Slack-C-Myc) with 3×HA-tagged Na$_V$1.6's termini (3×HA-Na$_V$1.6-N or 3×HA-Na$_V$1.6-C) in HEK293T cell lysates. The 3×HA tag was fused to the N-terminal region of Na$_V$1.6's fragments, and the Myc tag was fused to the C-terminal region of Slack's fragment.

The online version of this article includes the following source data for figure 5:

**Source data 1.** Numerical data for *Figure 5*.

supporting that disrupting the Slack-Na$_V$1.6 interaction can indeed reduce current amplitudes of Slack mutant variants, which may protect against seizures induced by Slack mutant variants.

We next induced an in vivo epilepsy model by introducing a Slack G269S variant into C57BL/6N mice using adeno-associated virus (AAV) injection to mimic the human Slack mutation G288S (*Rizzo et al., 2016*). Specifically, we delivered stereotactic injections of AAV9 containing expression cassettes for Slack$^{G269S}$ (or GFP negative controls) into the hippocampal CA1 region of 3-week-old C57BL/6N mice (*Figure 7C and D*). At 3–5-week intervals after AAV injection, we quantified the seizure susceptibility of mice upon the induction of a classic kainic acid (KA) model of temporal lobe epilepsy (*Nadler, 1981*; *Lévesque and Avoli, 2013*). In this model, seizures with stage IV or higher (as defined by a modified Racine, Pinal, and Rovner scale; *Pinel and Rovner, 1978*) are induced in rodents by intraperitoneal administration of 28 mg/kg KA.

We assessed a time course of KA-induced seizure stages at 10 min intervals and found that viral expression of Slack$^{G269S}$ resulted in faster seizure progression in mice compared to control GFP expression (*Figure 7E*). We calculated the total seizure score per mouse to assess seizure severity (*Kim et al., 2021*). Slack$^{G269S}$-expressing mice showed significantly higher seizure severity than GFP-expressing mice (*Figure 7F*). The percentage of mice with stage VI–IX seizures also increased, from 9.1% in GFP-expressing control mice to 58.3% in the Slack$^{G269S}$-expressing mice (*Figure 7G*). These results support that viral expression of Slack$^{G269S}$ significantly increases seizure susceptibility in mice.

To evaluate the potential therapeutic effects of disrupting the Slack-Na$_V$1.6 interaction, we delivered two AAV9s (one for Slack$^{G269S}$ and one for Slack's C-terminus [residues 326–1238]) into the CA1

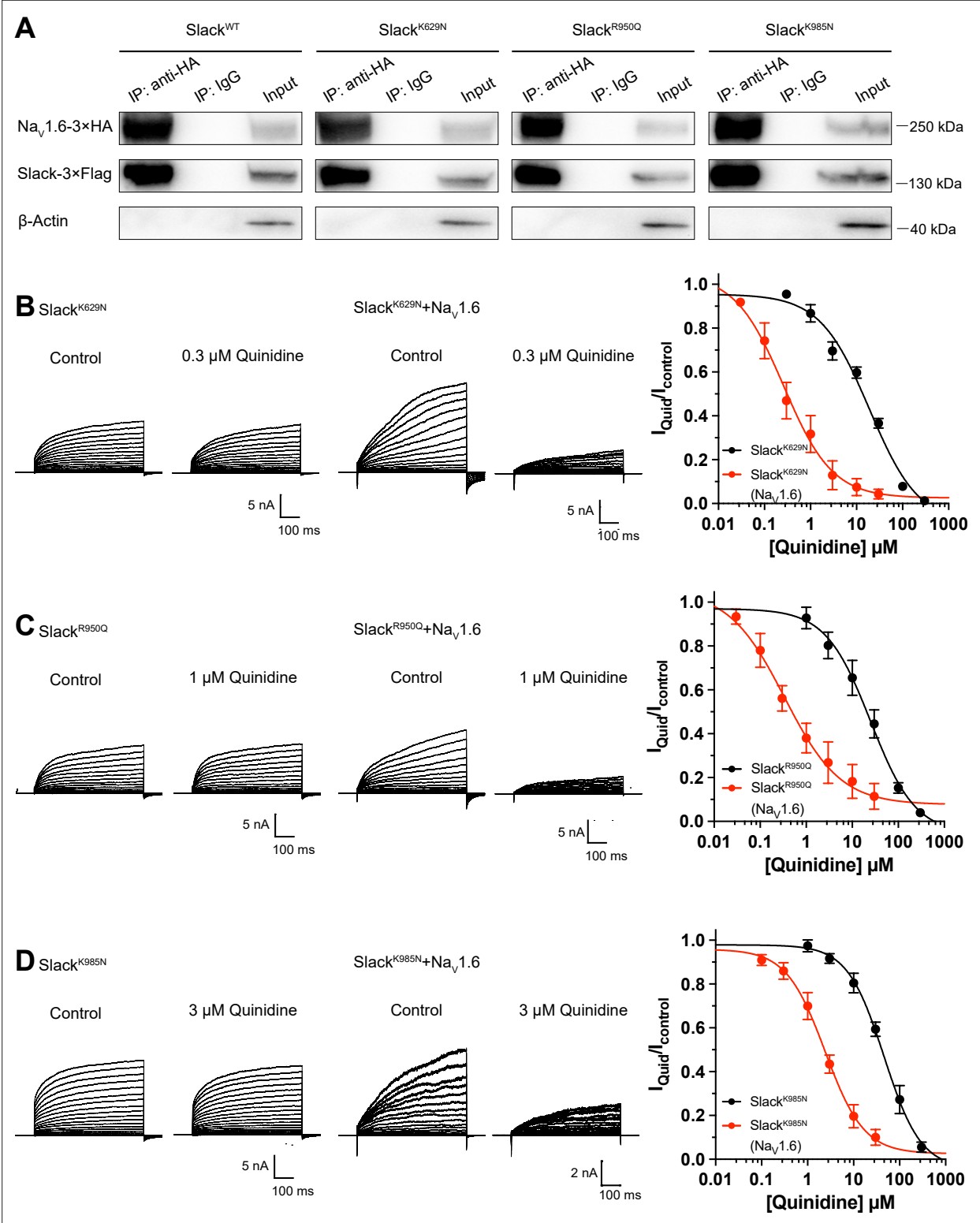

**Figure 6.** Na$_V$1.6 sensitizes epilepsy-related Slack mutant variants to quinidine blockade. (**A**) Co-IP of 3×Flag-tagged Slack or its mutations (Slack-3×Flag) with 3×HA-tagged Na$_V$1.6 (Na$_V$1.6–3×HA) in HEK293T cell lysates. The tags were all fused to the C-terminal region of wild-type or mutant ion channels. (**B–D**) The sensitivity of Slack mutant variants (Slack$^{K629N}$ [**B**], Slack$^{R950Q}$ [**C**], and Slack$^{K985N}$ [**D**]) to quinidine blockade upon expression of Slack mutant variants alone and co-expression of Slack mutant variants with Na$_V$1.6. Left: example current traces recorded from HEK293 cells expressing Slack mutant variants alone and co-expressing Slack mutant variants with Na$_V$1.6, before and after application of the indicated concentrations of quinidine.

*Figure 6 continued on next page*

*Figure 6 continued*

Right: the concentration–response curves for blocking of Slack mutant variants by quinidine upon expression of Slack mutant variants alone (n = 8 for Slack$^{K629N}$, n = 7 for Slack$^{R950Q}$, and n = 5 for Slack$^{K985N}$) and co-expression of Slack mutant variants with Na$_V$1.6 (n = 8 for Slack$^{K629N}$ upon co-expression with Na$_V$1.6, n = 5 for Slack$^{R950Q}$ upon co-expression with Na$_V$1.6, and n = 7 for Slack$^{K985N}$ upon co-expression with Na$_V$1.6). Please refer to **Supplementary file 1d** for IC$_{50}$ values.

The online version of this article includes the following source data and figure supplement(s) for figure 6:

**Source data 1.** Numerical data for *Figure 6* and *Figure 6—figure supplement 1*.

**Figure supplement 1.** The Na$^+$-mediated currents coupling of Slack mutant variants and Na$_V$1.6 upon co-expression in HEK293 cells.

region of mice (*Figure 7D*). Viral expression of Slack's C-terminus in Slack$^{G269S}$-expressing mice significantly decreased seizure progression, seizure severity, and the percentage of mice experiencing stage VI–IX seizures (*Figure 7E–G*). These results support that viral expression of Slack's C-terminus can prevent Slack$^{G269S}$-induced seizures in mice, thus showcasing that using Slack's C-terminus to disrupt the Slack-Na$_V$1.6 interaction is a promising therapeutic intervention to treat KCNT1-related epilepsy (*Figure 8*).

## Discussion

We here found that Na$_V$1.6's N- and C- termini bind to Slack's C-terminus and sensitize Slack to quinidine blockade via Na$_V$1.6-mediated transient sodium currents. These results suggest that the pharmacological blocking effects of a channel blocker are not exclusively mediated by the channel per se, but modulated by channel's interacting proteins. Moreover, we show that viral expression of Slack's C-terminus can rescue the increased seizure susceptibility and confer protection against Slack$^{G269S}$-induced seizures in mice.

At resting membrane potential, the intracellular sodium concentration ([Na$^+$]$_{in}$) in neurons (equal to ~10 mM) is too low to effectively activate Slack (K$_d$ of 66 mM) (*Kameyama et al., 1984*; *Zhang et al., 2010*). Slack is functionally coupled to sodium influx, which is known to be mediated by ion channels and receptors, including Na$_V$, AMPARs, and NMDARs (*Budelli et al., 2009*; *Hage and Salkoff, 2012*; *Nanou et al., 2008*; *Ehinger et al., 2021*). Such Na$^+$ sources can provide both the membrane depolarization and the Na$^+$ entry known to be required for Slack activation, enabling Slack to contribute both to action potential repolarization during neuronal high-frequency firing (*Wallén et al., 2007*; *Markham et al., 2013*) and to regulating excitatory postsynaptic potential (EPSP) at postsynaptic neurons (*Nanou et al., 2008*). Our results support that Slack and Na$_V$1.6 form a channel complex, while also implying that Na$_V$1.6-mediated sodium influx increases the Na$^+$ concentration in the close vicinity of Slack to activate Slack. As a low-threshold Na$_V$ channel subtype, Na$_V$1.6 has been reported to dominate the initiation and propagation of action potentials in AIS in excitatory neurons (*Hu et al., 2009*; *Kole et al., 2008*; *Lazarov et al., 2018*; *Katz et al., 2018*). The activation of Slack by Na$_V$1.6 at AIS has multiple impacts, including ensuring the timing of the fast-activated component of Slack currents, regulating the action potential amplitude, and apparently contributing to intrinsic neuronal excitability (*Kaczmarek, 2013*). The use of HEK cells and cultured primary cortical neurons in this study may not fully capture the complexity of native Slack-Na$_V$1.6 interaction. Future studies employing more intact systems, such as in vivo models, could provide a more comprehensive understanding of the physiological relevance of the native Slack-Na$_V$1.6 interaction.

An interesting question arises from our observation that Slack's sensitivity to quinidine blockade is enhanced by I$_{NaT}$ but not I$_{NaP}$: what can explain the distinct I$_{NaT}$ and I$_{NaP}$ contributions? We speculate that with physical modulation by Na$_V$1.6, I$_{NaT}$ may elicit a specific open conformation of Slack that brings its quinidine binding pocket into a high-affinity state, which could lead to a substantial increase in Slack's sensitivity to quinidine blockade. The possibility of this hypothetical open conformation is supported by previous reports of the presence of subconductance states detected in single-channel recordings of *Xenopus* oocytes expressing Slack channels (*Brown et al., 2008*), which implies that there are multiple open conformations of Slack. Although only one open conformation of Slack has been observed in cryo-EM, the gap between maximum conformational open probability (~1.0) and maximum functional open probability (~0.7) implies a subclass within this class of open channels (*Hite and MacKinnon, 2017*). Additionally, it is worth noting that blocking of I$_{NaP}$ by application of riluzole has its limitation on selectivity. Riluzole has been reported to bind to Slack channels with low affinity

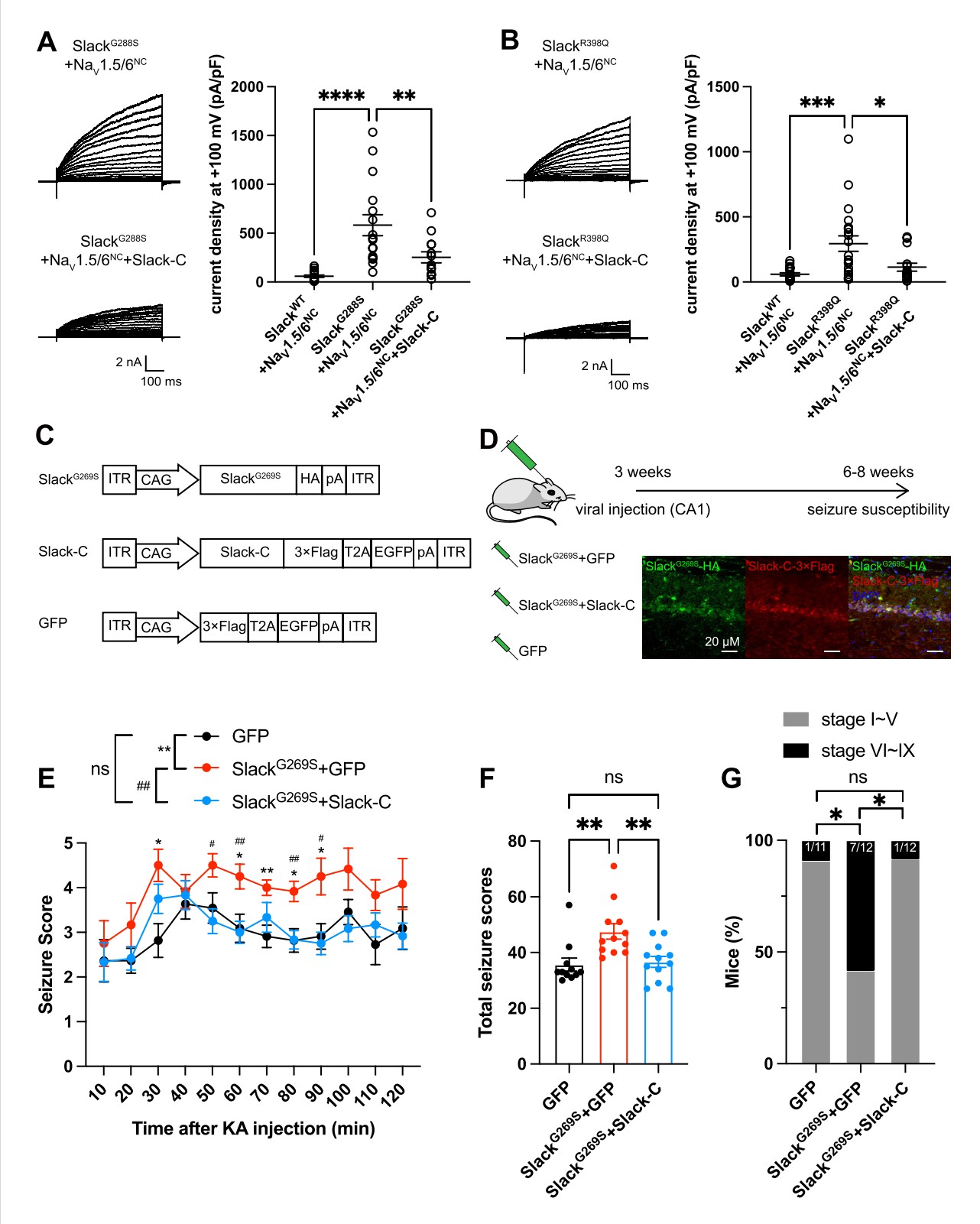

**Figure 7.** Viral expression of Slack's C-terminus prevents Slack[G269S]-induced seizures. (**A, B**) The current densities of Slack mutant variants (Slack[G288S] [**A**] and Slack[R398Q] [**B**]) upon co-expression with Na$_V$1.5/6[NC] in HEK293T cells were reduced by additional expression of Slack's C-terminus. Left: example current traces from HEK293T cells co-expressing Slack mutant variants and Na$_V$1.5/6[NC] or co-expressing Slack mutant variants, Na$_V$1.5/6[NC], and Slack's C-terminus. Right: summarized current densities at +100 mV. *p<0.05, **p<0.01, ***p<0.001; one-way ANOVA followed by Bonferroni's post hoc test.

*Figure 7 continued on next page*

*Figure 7 continued*

(**C**) Architecture for expression cassettes of AAVs. (**D**) Top: study design and timeline for the stereotactic injection model. Bottom: immunofluorescence of HA-tagged Slack$^{G269S}$ (green), 3×Flag-tagged Slack's C-terminus (red), and DAPI (blue) in hippocampal CA1 pyramidal cell layer at 5 wk after viral injection of Slack$^{G269S}$ with Slack's C-terminus into CA1 of mice. (**E**) Time course of kainic acid (KA)-induced seizure stage changes at 10 min intervals based on a modified Racine, Pinal, and Rovner scale (please refer to 'Methods' for further details). The number of mice used: 'GFP' control group (n = 11), 'Slack$^{G269S}$+GFP' group (n = 12), 'Slack$^{G269S}$+Slack-C' group (n = 12). 'GFP" vs. "Slack$^{G269S}$+GFP': $F_{(1,21)}$ = 10.48, p=0.0040, *p<0.05, **p<0.01; 'Slack$^{G269S}$+GFP' vs. 'Slack$^{G269S}$+Slack-C': $F_{(1,22)}$ = 10.30, p=0.0040, #p<0.05, ##p<0.01. 'GFP' vs. 'Slack$^{G269S}$+Slack-C': $F_{(1,21)}$ = 0.09574, p=0.7600. Repeated two-way ANOVA followed by Bonferroni's post hoc test. (**F**) Total seizure score per mouse over the 2 hr after KA injection of these three groups. *p<0.05, **p<0.01; one-way ANOVA followed by Bonferroni's post hoc test. (**G**) The percentage of mice with stage VI–IX seizures over the 2 hr after KA injection in each group. *p<0.05; Fisher's exact test.

The online version of this article includes the following source data and figure supplement(s) for figure 7:

**Source data 1.** Numerical data for *Figure 7* and *Figure 7—figure supplement 1*.

**Figure supplement 1.** Heterozygous knockout of Na$_V$1.6 decreases the amplitude of slow afterhyperpolarization (AHP) in hippocampal CA1 pyramidal neurons.

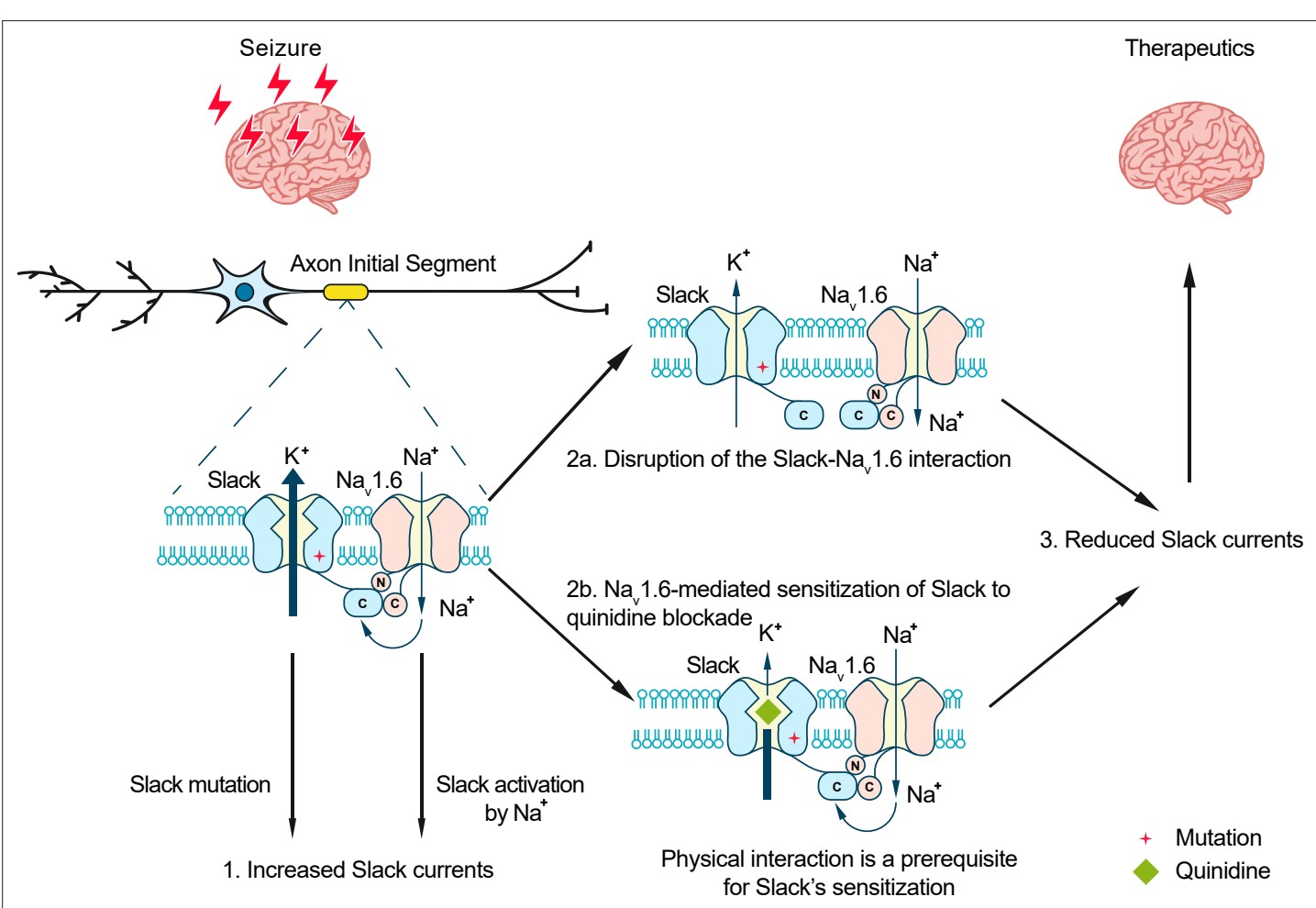

**Figure 8.** Model for protection against seizures by quinidine blockade or disruption of the Slack-Na$_V$1.6 interaction. The axon initial segments (AIS)-localized Slack-Na$_V$1.6 complex provides a plausible explanation for Na$_V$1.6-mediated sensitization of Slack to quinidine blockade. This sensitization requires physical interaction between Na$_V$1.6 and Slack and Na$_V$1.6-mediated sodium influx. Moreover, disruption of the Slack-Na$_V$1.6 interaction by expression of Slack's C-terminus reduces the amount of Na$^+$ in the close vicinity of Slack and thus reduces Slack currents. Therefore, the blocking of Slack by quinidine and disruption of the Slack-Na$_V$1.6 interaction can both convert the gain-of-function phenotypes of epilepsy-related Slack mutant variants and protect against seizures.

(*Biton et al., 2012*). Therefore, the dual binding of riluzole to both Slack and Na$_V$1.6 may stabilize or inhibit the Slack-Na$_V$1.6 interaction, consequently influencing the sensitization of Slack to quinidine blockade.

As previously mentioned, GOF Slack mutant variants have been linked to a broad spectrum of epileptic disorders that are accompanied by intellectual disabilities and both psychomotor and developmental defects (*Cole et al., 2021*; *Gertler et al., 1993*). Given that many patients are refractory or nonresponsive to conventional anticonvulsants (*McTague et al., 2018*; *Gertler et al., 1993*; *Fitzgerald et al., 2019*), and considering the limited success of quinidine in clinical treatment of KCNT1-related epilepsy, inhibitors targeting Slack are needed urgently (*Cole et al., 2021*). Several small-molecule inhibitors against Slack have been reported, providing informative starting points for drug development efforts with KCNT1-related epilepsy (*Cole et al., 2020*; *Griffin et al., 2021*; *Spitznagel et al., 2020*). However, our discovery of the Slack-Na$_V$1.6 complex challenges the traditional view that Slack acts as an isolated target in KCNT1-related epilepsy (*Cole et al., 2021*). Indeed, our study supports that co-expression of Slack and Na$_V$1.6 in heterologous cell models should be performed when analyzing clinically relevant Slack mutations and when screening anti-epileptic drugs for use in treating KCNT1-related disorders.

Genotype–phenotype analysis has shown that ADSHE-related mutations are clustered in the regulator of conductance of K$^+$ (RCK2) domain; while EIMFS-related mutations do not show a particular pattern of distribution (*Bonardi et al., 2021*). All functionally tested Slack mutant variants show GOF phenotypes, with increased Slack currents (*Barcia et al., 2012*; *Bonardi et al., 2021*; *Cole et al., 2021*). Further, the epilepsy-related Slack mutant variants confer their GOF phenotypes through two molecular mechanisms: increasing maximal channel open probability (P$_{max}$) or increasing sodium sensitivity (K$_d$) of Slack (*Tang et al., 2016*). Both P$_{max}$ and K$_d$ are highly sensitive to [Na$^+$]$_{in}$; these are respectively analogous to the efficacy and potency of [Na$^+$]$_{in}$ on Slack currents (*Tang et al., 2016*). Notably, several Slack mutant variants show GOF phenotypes only at high [Na$^+$]$_{in}$ (e.g., 80 mM) (*Tang et al., 2016*). These results indicate that the GOF phenotype of epilepsy-related Slack mutant variants is aggravated by high [Na$^+$]$_{in}$. Our discovery of functional coupling between Na$_V$1.6 and Slack presents a plausible basis for how Na$_V$1.6-mediated sodium influx can increase [Na$^+$]$_{in}$ and thus apparently aggravate the GOF phenotype of Slack mutant variants. Therefore, it makes sense that disruption of the Slack-Na$_V$1.6 interaction by overexpressing Slack's C-terminus reduces the current amplitudes of GOF Slack mutant variants (*Figure 7A and B*). Our successful demonstration that viral expression of Slack's C-terminus prevents epilepsy-related Slack$^{G269S}$-induced seizures in mice (*Figure 7E–G*) warrants further translational evaluation for developing therapeutic interventions to treat KCNT1-related epilepsy.

# Methods

## Animals

C57BL/6 mice were purchased from Charles River Laboratories. Na$_V$1.6 knockout C3HeB/FeJ mice were generous gifts from Professor Yousheng Shu at Fudan University. All animals were housed on a 12 hr light/dark cycle with ad libitum access to food and water. All procedures related to animal care and treatment were approved by the Peking University Institutional Animal Care and Use Committee (no. LA2020096) and met the guidelines of the National Institute of Health Guide for the Care and Use of Laboratory Animals. Each effort was made to minimize animal suffering and the number of animals used. The experiments were blind to viral treatment condition during behavioral testing.

## Antibodies and reagents

Commercial antibodies used were anti-AnkG (Santa Cruz), anti-Slack (NeuroMab), anti-Na$_V$1.2 (Alomone), anti-Na$_V$1.6 (Alomone), anti-HA (Abbkine), anti-Flag (Abbkine), anti-β-actin (Biodragon), HRP goat anti-mouse IgG LCS (Abbkine), HRP mouse anti-rabbit IgG LCS (Abbkine), Alexa Fluor 488-AffinityPure Fab Fragment donkey anti-rabbit IgG (Jackson), and Alexa Fluor 594 donkey anti-mouse IgG (Yeason). GPCR Extraction Reagent was from Pierce, NP40 lysis buffer was from Beyotime, protease inhibitor mixture cocktail was from Roche Applied Science, rabbit IgG and mouse IgG were from Santa Cruz, and Protein G Dynabeads were from Invitrogen. Tetrodotoxin was from Absin

Bioscience. Quinidine was from Macklin, and riluzole was from Meilunbio. All other reagents were purchased from Sigma-Aldrich.

## Molecular cloning

The ion channels used are Slack-B (*KCNT1*, RefSeq: NM_020822.3), Slick (*KCNT2*, RefSeq: NM_198503.5), $Na_V1.1$ (*SCN1A*, Ref Seq: NM_001165963.4), $Na_V1.2$ (*SCN2A*, Ref Seq: NM_001040142.2), $Na_V1.3$ (*SCN3A*, Ref Seq: NM_006922.4), $Na_V1.5$ (*SCN5A*, Ref Seq: NM_198056.3), $Na_V1.6$ (*SCN8A*, Ref Seq: NM_014191.4), GluA1 (*Gria1*, Ref Seq: XM_032913972.1), GluA2 (*Gria2*, Ref Seq: NM_017261.2), GluA3 (*GRIA3*, Ref Seq: NM_007325.5), and GluA4 (*GRIA4*, Ref Seq: NM_000829.4). Human Slack-B and $Na_V1.6$ were subcloned into the modified pcDNA3.1(+) vector using Gibson assembly. All mutations and chimeras of ion channels were also constructed using Gibson assembly. For GST pull-down assay, the segments of Slack and $Na_V1.6$ were subcloned into pCDNA3.1(+) and pGEX-4T-1 vector, respectively. For FRET experiments, mVenus-tag was fused to the C-terminus of $Na_V1.6$ sequence, and mTFP1-tag was also fused to the C-terminus of Slack sequence.

## Immunoprecipitation

The brain tissues or HEK293T cells co-expressing full-length or fragments of Slack and $Na_V1.6$ were homogenized and lysed in GPCR Extraction Reagent (Pierce) with cocktail for 30 min at 4°C. The homogenate was centrifuged for 20 min at $16,000 \times g$ and 4°C to remove cell debris and then supernatant was incubated with 5 µg Slack antibody (NeuroMab) or Nav1.6 antibody (Alomone) for 12 hr at 4°C with constant rotation. Then, 40 µL of protein G Dynabeads (Invitrogen) was added and the incubation was continued until the next day. Beads were then washed three times with NP40 lysis. Between washes, the beads were collected by DynaMag. The remaining proteins were eluted from the beads by resuspending the beads in 1×SDS-PAGE loading buffer and incubating for 30 min at 37°C. The resultant materials from immunoprecipitation or lysates were then subjected to western blot analysis. And the co-immunoprecipitation assays for Slack and $Na_V1.6$ were independently replicated three times, with biological replicates.

## Western blot analysis

Proteins suspended in 1× SDS-PAGE loading buffer were denatured for 30 min at 37°C. Then proteins were loaded on 6% or 8% sodium dodecyl sulfate–polyacrylamide gel electrophoresis and transferred onto nitrocellulose filter membrane (PALL). Nonspecific binding sites were blocked with Tris-buffered saline-Tween (0.02 M Tris, 0.137 M NaCl, and 0.1% Tween 20) containing 5% non-fat dried milk. Subsequently, proteins of interest were probed with primary antibodies for overnight at 4°C. After incubation with a secondary antibody, immunoreactive bands were visualized using HRP Substrate Peroxide Solution (Millipore) according to the manufacturer's recommendation.

## GST pull-down assay

Plasmids encoding GST-fused $Na_V1.6$ segments were transformed into BL21(DE3). After expressing recombinant proteins induced by overnight application of isopropyl-1-thio-β-D-galactopyranoside (IPTG) (0.1 mM) at 25°C, the bacteria were collected, lysed, and incubated with GSH beads using BeaverBeads GSH kit (Beaver) according to the manufacturer's instruction.

Plasmids encoding KCNT1 channel or HA-tagged KCNT1 segments were transfected into HEK293T cells using Lipofectamine 2000 (Invitrogen). Then, 40 hr after transfection, cells were lysed in NP40 lysis buffer with inhibitor cocktail, and then centrifuged at 4°C, $15,000 \times g$ for 20 min. The supernatants were incubated with protein-bound beads. The protein-bound beads were washed by washing buffer (50 mM Tris–HCl pH = 7.4, 120 mM NaCl, 2 mM EGTA, 2 mM DTT, 0.1% Triton X-100, and 0.2% Tween-20) for 5 min three times and then denatured with 1× SDS-PAGE loading buffer and incubating for 30 min at 37°C. The resultant materials were subjected to western blot analysis.

## Cell culture

The human embryonic kidney cells (HEK293 and HEK293T), obtained originally from ATCC, were authenticated via Short Tandem Repeat (STR) profiling and tested negative for mycoplasma contamination. The HEK cells were maintained in Dulbecco's Modified Eagle Medium (DMEM, Gibco) supplemented with 15% fetal bovine serum (FBS, PAN-Biotech) at 37°C and 5% $CO_2$. Primary cortical neurons

were prepared from either sex of postnatal (P0-P1) homozygous $Na_V1.6$ knockout C3HeB/FeJ mice and the wild-type littermate controls. After the mice were decapitated, the cortices were removed and separated from the meninges and surrounding tissue. Tissues were digested in 2 mg/mL Papain (Aladdin) containing 2 µg/mL DNase I (Psaitong) for 30 min followed by centrifugation and resuspension. Subsequently, the cells were plated on poly-D-lysine (0.05 mg/mL) pre-coated glass coverslips in plating medium (DMEM containing 15% FBS), at a density of $4 \times 10^5$ cells/mL, and cultivated at 37°C and 5% $CO_2$ in a humidified incubator. Then, 5 hr after plating, the medium was replaced with Neurobasal Plus medium (Invitrogen) containing 2% v/v B-27 supplement (Invitrogen), 2 mM GlutaMAX (Invitrogen), and 50 U/mL penicillin and streptomycin (Life Technologies). The primary neurons were grown 6–10 d before electrophysiological recordings with half of the media replaced every 3 d.

## Voltage-clamp recordings

The plasmids expressing full-length or fragments of Slack and $Na_V$ channels (excluding full-length $Na_V1.6$) were co-transfected into HEK293 cells using Lipofectamine 2000 (Invitrogen). To co-express Slack with $Na_V1.6$, the plasmid expressing Slack was transfected into a stable HEK293 cell line expressing $Na_V1.6$. 18–36 hr after transfection, voltage-clamp recordings were obtained using a HEKA EPC-10 patch-clamp amplifier (HEKA Electronic) and PatchMaster software (HEKA Electronic). For all whole-cell patch-clamp experiments in HEK cells except for data presented in *Figure 2—figure supplement 4D*, the extracellular recording solution contained (in mM) 140 NaCl, 3 KCl, 1 $CaCl_2$, 1 MgCl2, 10 glucose, 10 HEPES, and 1 tetraethylammonium chloride (310 mOsm/L, pH 7.30 with NaOH). The recording pipette intracellular solution (5 mM Na) contained (in mM) 100 K-gluconate, 30 KCl, 15 choline-Cl, 5 NaCl, 10 glucose, 5 EGTA, and 10 HEPES (300 mOsm/L, pH 7.30 with KOH). For data presented in *Figure 2—figure supplement 4D*, the extracellular recording solution remains the same as above, the pipette intracellular solution contained (in mM) 140 CsF, 10 NaCl, 5 EGTA, and 10 HEPES (pH 7.30 with NaOH), indicating that the unusual right shifts (15–20 mV) in voltage dependence of $Na_V1.6$ were induced by components in pipette solution, not recording system errors. For primary cortical neurons, intracellular solution (0 mM Na) was used to prevent the activation of sodium-activated potassium channels by basal intracellular sodium ions. NaCl in the intracellular solution was replaced with choline chloride in an equimolar concentration. For inside-out patch-clamps, the bath solution contained (in mM) 140 NaCl, 1 EDTA, 10 HEPES and 2 $MgCl_2$ (310 mOsm/L, pH 7.30 with NaOH). Pipette solution contained (in mM): 130 KCl, 1 EDTA, 10 HEPES, and 2 $MgCl_2$ (300 mOsm/L, pH 7.30 with KOH). The pipettes were fabricated by a DMZ Universal Electrode puller (Zeitz Instruments) using borosilicate glass, with a resistance of 1.5–3.5 MΩ for whole-cell patch-clamp recordings and 8.0–10.0 MΩ for inside-out patch-clamp recordings. All experiments were performed at room temperature. The cells were exposed to the bath solution with quinidine for about 1 min before applying voltage protocols. The concentration–response curves were fitted to four-parameter Hill equation:

$$Y = Bottom + (Top - Bottom) / \left(1 + 10^{(lgIC_{50}-X)*H}\right) \tag{1}$$

where $Y$ is the value of $I_{Quinidine}/I_{Control}$, Top is the maximum response, Bottom is the minimum response, $X$ is the $lg$ of concentration, $IC_{50}$ is the drug concentration producing the half-maximum response, and $H$ is the Hill coefficient. Significance of fitted $IC_{50}$ values compared to control was analyzed using extra sum-of-squares $F$-test.

For data presented in *Figure 2—figure supplement 4*, cells were excluded from analysis if series resistance >5 MΩ and series resistance compensation was set to 70–90%.

The time constants ($\tau$) of activation were fitted with a single exponential equation:

$$I(t) = Offset + A * exp\left(\frac{-t}{\tau}\right) \tag{2}$$

where $I$ is the current amplitude, $t$ is time, offset represents the asymptote of the fit, and $A$ represents the amplitude for the activation or inactivation.

Steady-state fast inactivation (I–V) and conductance–voltage (G–V) relationships were fitting with Boltzmann equations:

$$I/I_{max} = 1/\left(1 + exp\left(\frac{V_m - V_{1/2}}{k}\right)\right) \qquad (3)$$

$$G/G_{max} = 1/\left(1 + exp\left(\frac{V_m - V_{1/2}}{-k}\right)\right) \qquad (4)$$

$$G = I/\left(V_m - E_{Na}\right) \qquad (5)$$

where $I$ is the peak current, $G$ is conductance, $V_m$ is the stimulus potential, $V_{1/2}$ is the midpoint voltage, $E_{Na}$ is the equilibrium potential, and $k$ is the slope factor. Significance of fitted $V_{1/2}$ compared to control was analyzed using extra sum-of-squares $F$-test.

Recovery from fast inactivation data were fitted with a single exponential equation:

$$I/I_{max} = A * \left(1 - exp\left(\frac{-t}{\tau}\right)\right) \qquad (6)$$

where $I$ is the peak current of test pulse, $I_{max}$ is the peak current of first pulse, $A$ is the proportional coefficient, $t$ is the delay time between the two pulses, and $\tau$ is the time constant of recovery from fast inactivation.

The activation time constants of NaV channels were fitted with a single exponential equation:

$$I = B + A * exp\left(\frac{t}{\tau}\right) \qquad (7)$$

where $I$ is the current amplitude, $A$ and $B$ are the partition coefficients, $t$ is the time, and $\tau$ is the time constant of activation.

## Acute slice preparation and current-clamp recordings

Horizontal slices containing hippocampus were obtained from 6- to 8-week-old heterozygous $Na_V1.6$ knockout ($Scn8a^{+/-}$) C3HeB/FeJ mice and the wild-type littermate controls. In brief, animals were anesthetized and perfused intracardially with ice-cold modified 'cutting solution' containing (in mM) 110 choline chloride, 2.5 KCl, 0.5 $CaCl_2$, 7 $MgCl_2$, 25 $NaHCO_3$, 1.25 $NaH_2PO_4$, and 10 glucose; bubbled continuously with 95%$O_2$/5%$CO_2$ to maintain PH at 7.2. The brain was then removed and submerged in ice-cold 'cutting solution'. Next, the brain was cut into 300 μm slices with a vibratome (WPI). Slices were incubated in oxygenated (95% $O_2$ and 5% $CO_2$) 'recording solution' containing (in mM) 125 NaCl, 2.5 KCl, 2 $CaCl_2$, 2 $MgCl_2$, 25 $NaHCO_3$, 1.25 $NaH_2PO_4$, 10 glucose (315 mOsm/L, PH 7.4, 37°C) for 30 min, and stored at room temperature.

Slices were subsequently transferred to a submerged chamber containing 'recording solution' maintained at 34–36°C. Whole-cell recordings were obtained from hippocampal CA1 neurons under a ×60 water-immersion objective of an Olympus BX51WI microscope (Olympus). Pipettes had resistances of 5–8 MΩ. For current-clamp recordings, the external solution (unless otherwise noted) was supplemented with 0.05 mM (2R)-amino-5-phosphonovaleric acid (APV), 0.01 mM 6-cyano-7-nitro-q uinoxaline-2,3-dione, 0.01 mM bicuculline, and 0.001 mM CGP 55845, and internal pipette solution containing (in mM) 118 $KMeSO_4$, 15 KCl, 10 HEPES, 2 $MgCl_2$, 0.2 EGTA, and 4 $Na_2ATP$, 0.3 Tris-GTP, 14 Tris-phosphocreatinin (pH 7.3 with KOH).

In our whole-cell current-clamp recordings with an Axon 700B amplifier (Molecular Devices), we initially applied a 100 ms, 20 pA test pulse to the recording neurons right after breaking into the whole-cell configuration. A fast-rising component and a slow-rising component of voltage response were clearly visible. Then we zoomed into the fast-rising component of the voltage responses and turned up the pipette capacitance neutralization slowly to shorten the rise time of the fast-rising component until the oscillations of voltage responses appeared. Subsequently, we decreased the capacitance compensation just until the oscillations disappeared. For bridge balance of current-clamp recordings, we increased the value of bridge balance slowly until the fast component of the voltage response disappeared, and the slow-rising component appeared to rise directly from the baseline. Series resistance and pipette capacitance were compensated using the bridge balance and pipette capacitance neutralization options in the Multiclamp 700B command software (Molecular Devices). The bridge balance value was between 20 and 30 MΩ and the pipette capacitance neutralization

value was between 3 and 5 pF. For in vitro experiments, the cells were selected by criteria based on hippocampal CA1 cell morphology and electrophysiological properties in the slices. Electrophysiological recordings were made using a Multiclamp 700B amplifier (Molecular Devices). Recordings were filtered at 10 kHz and sampled at 50 kHz. Data were acquired and analyzed using pClamp10.0 (Molecular Devices). Series resistance was in the order of 10–30 MΩ and was approximately 60–80% compensated. Recordings were discarded if the series resistance increased by more than 20% during the time course of the recordings.

## Fluorescence imaging and FRET quantification

The spectroscopic imaging was built upon a Nikon TE2000-U microscope. The excitation light was generated by an Ar laser. The fluorescent protein mVenus fused to $Na_V1.6$ and mTFP1 fused to Slack were excited by laser line at 500 and 400–440 nm, respectively. The duration of light exposure was controlled by a computer-driven mechanical shutter (Uniblitz). A spectrograph (Acton SpectraPro 2150i) was used in conjunction with a charge coupled device (CCD) camera (Roper Cascade 128B). In this recording mode, two filter cubes (Chroma) were used to collect spectroscopic images from each cell (excitation, dichroic): cube I, D436/20, 455dclp; cube II, HQ500/20, Q515lp. No emission filter was used in these cubes. Under the experimental conditions, auto fluorescence from untransfected cells was negligible. Fluorescence imaging and analysis were done using the MetaMorph software (Universal Imaging). User-designed macros were used for automatic collection of the bright-field cell image, the fluorescence cell image, and the spectroscopic image. Emission spectra were collected from the plasma membrane of the cell by positioning the spectrograph slit across a cell and recording the fluorescence intensity at the position corresponding to the membrane region (*Figure 2E*, dotted lines in red); the same slit position applied to both the spectrum taken with the mTFP1 excitation and the spectrum taken with the mVenus excitation. Using this approach, the spectral and positional information is well preserved, thus allowing reliable quantification of FRET efficiency specifically from the cell membrane. Spectra were corrected for background light, which was estimated from the blank region of the same image.

FRET data was quantified in two ways. First, the FRET ratio was calculated from the increase in mVenus emission due to energy transfer as described in the previous study. (*Qiu et al., 2005*) Briefly, mTFP1 emission was separated from mVenus emission by fitting of standard spectra acquired from cells expressing only mVenus or mTFP1. The fraction of mVenus-tagged molecules that are associated with mTFP1-tagged molecules, *Ab*, is calculated as

$$Ab = 1/\left(1 + K_D/\left[Dfree\right]\right) \tag{8}$$

where $K_D$ is the dissociation constant and [Dfree] is the concentration of free donor molecules. Note that

$$FRETRatio = 1 + Ab * \left(FRETRatio_{max} - 1\right) \tag{9}$$

Regression analysis was used to estimate *Ab* in individual cells. From each cell, the FRET ratio$_{exp}$ was experimentally determined. The predicted *Ab* value was then computed by adjusting two parameters, FRET Ratio$_{max}$ and apparent $K_D$. *Ab* was in turn used to give a predicted FRET ratio$_{predicted}$. By minimizing the squared errors (FRET ratio$_{exp}$ – FRET ratio$_{predicted}$) (*Yuan et al., 2003*), $K_D$ was determined.

Second, apparent FRET efficiency was also calculated from the enhancement of mVenus fluorescence emission due to energy transfer (*Qiu et al., 2005*; *Cheng et al., 2007*; *Zou et al., 2017*; *Zheng et al., 2002*) using a method as previously described. (*Yang et al., 2010*) Briefly, Ratio $A_0$ and Ratio *A* were measured to calculate FRET efficiency. Ratio $A_0$ represents the ratio between tetramethylrhodamine maleimide emission intensities (in the absence of fluorescein maleimide) upon excitation at the donor and acceptor excitation wavelengths (*Zheng et al., 2002*; *Yang et al., 2010*; *Erickson et al., 2001*), and was calculated in the present study at the mVenus peak emission wavelength. A particular advantage of quantifying Ratio $A_0$ for FRET measurement is that changes in fluorescence intensity caused by many experimental factors can be canceled out by the ratiometric measurement. A similar ratio, termed Ratio *A*, was determined in the presence of mTFP1 in the same way as Ratio $A_0$. If FRET occurred, the Ratio *A* value should be higher than Ratio $A_0$; the difference between Ratio

*A* and Ratio $A_0$ was directly proportional to the FRET efficiency by the factor of extinction coefficient ratio of mTFP1 and mVenus (*Zheng et al., 2002*; *Yang et al., 2010*; *Erickson et al., 2001*).

## Immunostaining

After deep anesthesia with sodium pentobarbital, mice were sacrificed by perfusion with 0.5% para-formaldehyde and 0.5% sucrose (wt/vol) in 0.1 M phosphate buffer (pH 7.4). The brain was removed and post-fixed in the same fixative for 2 hr, and subsequently immersed in 30% sucrose in 0.1 M phosphate buffer for 48 hr. Cryostat coronal sections (20 μm) were obtained using a freezing microtome (Leica). The sections were rinsed in 0.01 M phosphate-buffered saline (PBS, pH 7.4), permeabilized in 0.5% Triton X-100 in PBS for 30 min, and incubated in a blocking solution (5% BSA, 0.1% Triton X-100 in PBS, vol/vol) at 20–25°C for 2 hr, followed by overnight incubation at 4°C with primary antibody to AnkG (1:100, Santa Cruz, sc-31778), Slack (1:100, NeuroMab, 73-051), $Na_V1.2$ (1:200, Alomone, ASC-002), $Na_V1.6$ (1:200, Alomone, ASC-006), Flag (1:500, Abbkine, ABT2010), and HA (1:500, Abbkine, ABT2040) in blocking solution. After a complete wash in PBS, the sections were incubated in Alexa 488-conjugated donkey anti-rabbit IgG and Alexa 594 donkey anti-mouse IgG in blocking solution at 20–25°C for 2 hr. The sections were subsequently washed and rinsed in DAPI solution. Images were taken in the linear range of the photomultiplier with a laser scanning confocal microscope (ZEISS LSM 510 META NLO).

## Adeno-associated virus construction and injection

The adeno-associated viruses (AAVs) and the negative GFP control were from Shanghai GeneChem. Co., Ltd. The full-length $Slack_{G269S}$ sequence (1-1238aa) was ligated into modified CV232 (CAG-MCS-HA-Poly A) adeno-associated viral vector. The Slack's C-terminus sequence (326-1238aa) and the negative control were ligated into GV634 (CAG-MCS-3×Flag-T2A-EGFP-SV40-Poly A) adeno-associated viral vector. The viruses (>$10^{11}$ TU/mL) were used in this study.

For dorsal CA1 viral injection, C57BL/6J mice aged 3 wk were anesthetized with isoflurane and placed in a stereotaxic apparatus (RWD Life Science Co., Ltd). Using a 5 μL micro syringe (Hamilton) with a 30-gauge needle (RWD Life Science Co., Ltd), 600 nL of the viruses was delivered at 10 nL/min by a micro-syringe pump (RWD Life Science Co., Ltd) at the following site in each of the bilateral CA1 regions, using the stereotaxic coordinates: 2.5 mm (anterior–posterior) from bregma, 2 mm (mediolateral), ±1.5 mm (dorsal–ventral) (*Chai et al., 2021*). The syringe was left in place for 5 min after each injection and withdrawn slowly. The exposed skin was closed by surgical sutures and returned to home cage for recovery. All the experiments were conducted after at least 3 wk of recovery. All the mice were sacrificed after experiments to confirm the injection sites and the viral trans-infection effects by checking EGFP under a fluorescence microscope (ZEISS LSM 510 META NLO).

## Kainic acid-induced status epilepticus

KA (Sigma-Aldrich) was intraperitoneally administered to produce seizures with stage IV or higher. The dose of kainic acid used was 28 mg/kg for mice (6–8 wk) (*Huang et al., 2009*; *He et al., 2004*). To assess epilepsy susceptibility, seizures were rated using a modified Racine, Pinal, and Rovner scale (*Pinel and Rovner, 1978*; *Racine, 1972*): (1) facial movements; (2) head nodding; (3) forelimb clonus; (4) dorsal extension (rearing); (5) loss of balance and falling; (6) repeated rearing and failing; (7) violent jumping and running; (8) stage 7 with periods of tonus; and (9) dead. Seizures was terminated 2 hr after onset with the use of sodium pentobarbital (30 mg/kg; Sigma-Aldrich).

## Statistical analysis

For in vitro experiments, the cells were evenly suspended and then randomly distributed in each well tested. For in vivo experiments, the animals were distributed into various treatment groups randomly. Statistical analyses were performed using GraphPad Prism 9 (GraphPad Software) and SPSS 26.0 software (SPSS Inc). Before statistical analysis, variation within each group of data and the assumptions of the tests were checked. Comparisons between two independent groups were made using unpaired Student's two-tailed *t*-test. Comparisons among nonlinear fitted values were made using extra sum-of-squares *F*-test. Comparisons among three or more groups were made using one- or two-way ANOVA followed by Bonferroni's post hoc test. No statistical methods were used to predetermine sample sizes but our sample sizes are similar to those reported previously in the field (*Liu et al.,*

*2017*; *Huang et al., 2012*). All experiments and analysis of data were performed in a blinded manner by investigators who were unaware of the genotype or manipulation. $*p<0.05$, $**p<0.01$, $***p<0.001$, $****p<0.0001$. All data are presented as mean ± SEM.

## Acknowledgements

This work was supported by Chinese National Programs for Brain Science and Brain-like intelligence technology no. 2021ZD0202102 to ZH; National Natural Science Foundation of China Grant (nos. 31871083 and 81371432 to ZH; nos. 32000674 to GD). We also thank Professor Yousheng Shu at Fudan University for providing the $Na_V1.6$-knockout mice.

## Additional information

### Funding

| Funder | Grant reference number | Author |
|---|---|---|
| National Natural Science Foundation of China-China Academy of General Technology Joint Fund for Basic Research | Nos. 31871083 and 81371432 | Zhuo Huang |
| National Natural Science Foundation of China | Nos. 32000674 | Guifang Duan |
| Chinese National Programs for Brain Science and Brain-like Intelligence Technology | No.2021ZD0202102 | Zhuo Huang |

The funders had no role in study design, data collection and interpretation, or the decision to submit the work for publication.

### Author contributions

Tian Yuan, Yuchen Jin, Data curation, Formal analysis, Validation, Investigation, Visualization, Methodology, Writing - original draft; Yifan Wang, Heng Zhang, Na Li, Xinyue Ma, Data curation, Formal analysis, Validation, Investigation, Visualization, Methodology; Hui Yang, Data curation, Formal analysis, Validation, Investigation, Visualization; Shuai Xu, Qian Chen, Data curation, Validation, Investigation, Visualization, Methodology; Huifang Song, Data curation, Formal analysis, Investigation, Methodology; Chao Peng, Formal analysis, Validation, Investigation; Ze Geng, Jie Dong, Validation, Investigation; Guifang Duan, Funding acquisition, Writing – review and editing; Qi Sun, Yang Yang, Writing – review and editing; Fan Yang, Resources, Writing – review and editing; Zhuo Huang, Conceptualization, Resources, Supervision, Funding acquisition, Project administration, Writing – review and editing

### Author ORCIDs

Xinyue Ma http://orcid.org/0000-0001-6114-1593
Zhuo Huang https://orcid.org/0000-0001-9198-4778

### Ethics

All procedures related to animal care and treatment were approved by the Peking University Institutional Animal Care and Use Committee (No. LA2020096) and met the guidelines of the National Institute of Health Guide for the Care and Use of Laboratory Animals. Each effort was made to minimize animal suffering and the number of animals used.

Reviewer #1 (Public Review): https://doi.org/10.7554/eLife.87559.4.sa1
Reviewer #2 (Public Review): https://doi.org/10.7554/eLife.87559.4.sa2
Reviewer #3 (Public Review): https://doi.org/10.7554/eLife.87559.4.sa3
Author Response https://doi.org/10.7554/eLife.87559.4.sa4

# Additional files

## Supplementary files

• Supplementary file 1. The IC50 values for quinidine sensitivity and kinetic characteristics of Slack and NaV1.x channels. (**a**) The sensitivity of Slack to quinidine blockade upon expression of Slack alone and co-expression of Slack with Na$_V$1.x. (**b**) The sensitivity of Na$_V$ channel subtypes to quinidine blockade upon expression of Na$_V$1.x alone and co-expression of Na$_V$1.x with Slack. (**c**) Biophysical characteristics of Na$_V$1.6 expressed alone and Na$_V$1.6 upon co-expression with Slack. (**d**) The sensitivity of Slack mutant variants to quinidine blockade upon expression of Slack mutant variants alone and co-expression of Slack mutant variants with Na$_V$1.6.

• MDAR checklist

## Data availability

All data generated or analyzed during this study are included in the manuscript and supporting file; Source Data files have been provided for figures.

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
