## [Editor Report · eLife assessment]

The authors report that an interaction between the sodium-activated potassium channel Slack and Na_V_1.6 sensitizes Slack to inhibition by quinidine. This is an **important** finding because it contributes to our understanding of how the antiseizure drug quinidine affects epilepsy syndromes arising from mutations in the Slack-encoding gene KCNT1. The results are largely **compelling** and the work will likely spark interest in further examining the proposed channel-channel interaction in neuronal cell membranes.

---

## [Referee Report · Reviewer #1 (Public Review)]

Despite numerous studies on quinidine therapies for epilepsies associated with GOF mutant variants of Slack, there is no consensus on its utility due to contradictory results. In this study Yuan et al. investigated the role of different sodium selective ion channels on the sensitization of Slack to quinidine block. The study employed electrophysiological approaches, FRET studies, genetically modified proteins and biochemistry to demonstrate that Nav1.6 N- and C-tail interacts with Slack's C-terminus and significantly increases Slack sensitivity to quinidine blockade in vitro and in vivo. This finding inspired the authors to investigate whether they could rescue Slack GOF mutant variants by simply disrupting the interaction between Slack and Nav1.6. They find that the isolated C-terminus of Slack can reduce the current amplitude of Slack GOF mutant variants co-expressed with Nav1.6 in HEK cells and prevent Slack induced seizures in mouse models of epilepsy. This study adds to the growing list of channels that are modulated by protein-protein interactions, and is of great value for future therapeutic strategies.

---

## [Referee Report · Reviewer #2 (Public Review)]

This is a very interesting paper about the coupling of Slack and Nav1.6 and the insight this brings to the effects of quinidine to treat some epilepsy syndromes.

Slack is a sodium-activated potassium channel that is important to hyperpolarization of neurons after an action potential. Slack is encoded by KNCT1 which has mutations in some epilepsy syndromes. These types of epilepsy are treated with quinidine but this is an atypical antiseizure drug, not used for other types of epilepsy. For sufficient sodium to activate Slack, Slack needs to be close to a channel that allows robust sodium entry, like Nav channels or AMPA receptors. but more mechanistic information is not available. Of particular interest to the authors is what allows quinidine to be effective in reducing Slack.

In the manuscript, the authors show that Nav, not AMPA receptors, are responsible for Slack's sensitization to quinidine blockade, at least in cultured neurons (HeK293, primary cortical neurons). Most of the paper focuses on the evidence that Nav1.6 promotes Slack sensitivity to quinidine.

---

## [Referee Report · Reviewer #3 (Public Review)]

Yuan et al., set out to examine the role of functional and structural interaction between Slack and NaVs on the Slack sensitivity to quinidine. Through pharmacological and genetic means they identify NaV1.6 as the privileged NaV isoform in sensitizing Slack to quinidine. Through biochemical assays, they then determine that the C-terminus of Slack physically interacts with the N- and C-termini of NaV1.6. Using the information gleaned from the in vitro experiments the authors then show that virally-mediated transduction of Slack's C-terminus lessens the extent of SlackG269S-induced seizures. These data uncover a previously unrecognized interaction between a sodium and a potassium channel, which contributes to the latter's sensitivity to quinidine.

---

## [Author Response]

The following is the authors’ response to the previous reviews.

We appreciate the constructive comments made by the editor and the reviewers. We have corrected errors and provided additional experimental data and analysis to address the latest criticisms raised by the reviewers and provided point-by-point response to the reviewers as below.

**Reviewer #1 (Recommendations For The Authors):**
I do acknowledge the work the authors put into this manuscript and I can accept the fact that the authors decided on a minimum of additional experiments. However, I would recommend the authors to be more concise by adding more information in the method and result sections about how they performed their experiments such as which Nav and AMPAR DNA constructs they used, the age of the mice, how long time they exposed the patches to quinidine, information on how many times they repeated their pull downs etc.

Answer: We thank the reviewer’s comments. we have incorporated the suggested modifications into our revised manuscript. Specifically, we have included detailed information on the NaV and AMPAR constructs in the Methods section. The age of the homozygous NaV1.6 knockout mice and the wild-type littermate controls is postnatal (P0-P1) (see in Results and Methods section). Prior to the application of step pulses, cells were subjected to the bath solution containing quinidine for approximately one minute (see in Methods section). Additionally, the co-immunoprecipitation assays for Slack and NaV1.6 were repeated three times (see in Methods section).

Minor detail in line 263: "...KCNT1 (Slack) have been identified to related to seizure..." I guess this should have been "...KCNT1 (Slack) have been identified and related to seizure..."?

Answer: We thank the reviewer for raising this point. We have corrected it in the revised manuscript.

Also, and again minor detail, I had a comment about the color coding in Fig 4 and by mistake, I added 4B, but I meant the use of colors in the entire figure, and mainly the use of colors in 4C, G and I.

Answer: We apologize for the confusion. We have changed the color coding of Figure 4 in the revised manuscript.

**Reviewer #2 (Recommendations For The Authors):**
While the paper is improved, several concerns do not seem to have been addressed. Some may have been missed because there is no response at all, but others may have been unclear because the response does not address the concern, but a related issue. Details are below.

Answer: We thank the reviewer for the criticisms. We have made changes of our manuscript to address the concerns.

Original issue:1. Remove the term in vivo.

Answer: We thank the reviewer for raising this point. In our experiments, although we did not conduct experiments directly in living organisms, our results demonstrated the coimmunoprecipitation of NaV1.6 with Slack in homogenates from mouse cortical and hippocampal tissues (Fig. 3C). This result may support that the interaction between Slack and NaV1.6 occurs in vivo.

New comment from reviewer:The argument to use the term in vivo is not well supported by what the authors have said. Just because tissues are used from an animal does not mean experiments were conducted in vivo. As the authors say, they did not conduct experiments in living organisms. Therefore the term in vivo should be avoided. This is a minor point.

Answer: We thank the reviewer for pointing this out. We have removed the term “in vivo” in the revised manuscript.

Original:1. Figure 1C Why does Nav1.2 have a small inward current before the large inward current in the inset?

Answer: We apologize for the confusion. We would like to clarify that the small inward current can be attributed to the current of membrane capacitance (slow capacitance or C-slow). The larger inward current is mediated by NaV1.2.

New comment:This is not well argued. Please note why the authors know the current is due to capacitance. Also, how do they know the larger current is due to NaV1.2? Please add that to the paper so readers know too.

Answer: We thank the reviewer’s comment. To provide a clearer representation of NaV1.2mediated currents in Fig. 1C, we have replaced the original example trace with a new one in which only one inward current is observed.

Original:The slope of the rising phase of the larger sodium current seems greater than Nav1.6 or Nav1.5. Was this examined?

Answer: Additionally, we did not compare the slope of the rising phase of NaV subtypes sodium currents but primarily focused on the current amplitudes.

New comment:This is not a strong answer. There seems to be an effect that the authors do not mention and evidently did not quantify that argues against their conclusion, which weakens the presentation.

Answer: We thank the reviewer’s comment. To assess the slope of the rising phase of NaV subtype currents, we compared the activation time constants of NaV1.2, NaV1.5, and NaV1.6 peak currents in HEK293 cells co-expressing NaV channel subtypes with Slack. The results have shown no significant differences (Author response image 1). We have included this analysis (see Fig. S9A) and the corresponding fitting equation (see in Methods section) in the revised manuscript.

**Author response image 1. sa4fig1:** The activation time constants of peak sodium currents in HEK293 cells co-expressing NaV1.2 (n = 6), NaV1.5 (n = 5), and NaV1.6 (n = 5) with Slack, respectively. ns, p > 0.05, one-way ANOVA followed by Bonferroni’s post hoc test.

Original:2D-E For Nav1.5 the sodium current is very large compared to Nav1.6. Is it possible the greater effect of quinidine for Nav1.6 is due to the lesser sodium current of Nav1.6?

Answer: We thank the reviewer for raising this point. We would like to clarify that our results indicate that transient sodium currents contribute to the sensitization of Slack to quinidine blockade (Fig. 2C,E). Therefore, it is unlikely that the greater effect observed for NaV1.6 in sensitizing Slack is due to its lower sodium currents.

New comment:I am not sure the question I was asking was clear. How can the authors discount the possibility that quinidine is more effective on NaV1.6 because the NaV1.6 current is relatively weak?

Answer: We thank the reviewer for raising this point. We have examined the sodium current amplitudes of NaV1.5, NaV1.5/1.6 chimeras, and NaV1.6 upon co-expression of NaV with Slack. Our analysis revealed that there are no significant differences between NaV1.5 and NaV1.5/6N, with both exhibiting much larger current amplitudes compared to NaV1.6 (Author response image 2), but only NaV1.5/6N replicates the effect of NaV1.6 in sensitizing Slack to quinidine blockade (Fig. 4H-I), suggesting the observed differences between NaV1.5 and NaV1.6 in sensitizing Slack are unlikely to be attributed to NaV1.6's lower sodium currents but may instead involve NaV1.6's Nterminus-induced physical interaction. We have included this analysis in the revised manuscript (see Fig. S9B).

**Author response image 2. sa4fig2:** Comparison of peak sodium current amplitudes of NaV1.5 (n = 9), NaV1.5/6NC (n = 13), NaV1.5/6N (n = 10), and NaV1.6 (n = 8) upon co-expressed with Slack in HEK293 cells. ns, p > 0.05, * p < 0.05, ** p < 0.01, *** p < 0.001, **** p < 0.0001; one-way ANOVA followed by Bonferroni’s post hoc test.

Original:The differences between WT and KO in G -H are hard to appreciate. Could quantification be shown? The text uses words like "block" but this is not clear from the figure. It seems that the replacement of Na+ with Li+ did not block the outward current or effect of quinidine.

Answer: We apologize for the confusion. We would like to clarify the methods used in this experiment. The lithium ion (Li+) is a much weaker activator of sodium-activated potassium channel Slack than sodium ion (Na+)1,2.

Zhang Z, Rosenhouse-Dantsker A, Tang QY, Noskov S, Logothetis DE. The RCK2 domain uses a coordination site present in Kir channels to confer sodium sensitivity to Slo2.2 channels. J Neurosci. Jun 2 2010;30(22):7554-62. doi:10.1523/JNEUROSCI.0525-10.2010Kaczmarek LK. Slack, Slick and Sodium-Activated Potassium Channels. ISRN Neurosci. Apr 18 2013;2013(2013)doi:10.1155/2013/354262Therefore, we replaced Na+ with Li+ in the bath solution to measure the current amplitudes of sodium-activated potassium currents (IKNa)3.Budelli G, Hage TA, Wei A, et al. Na+-activated K+ channels express a large delayed outward current in neurons during normal physiology. Nat Neurosci. Jun 2009;12(6):745-50. doi:10.1038/nn.2313

The following equation was used for quantification:IKna=OutwardcurrentamplitudesNa+basedbathsolution−OutwardcurrentamplitudesLi+basedbathsolution

Furthermore, the remaining IKNa after application of 3 μM quinidine in the bath solution was measured as the following:IKnawith3μMquinidine=OutwardcurrentamplitudesNa+basedbathsolutionwith3μMquinidine−OutwardcurrentamplitudesLi+basedbathsolutionwith3μMquinidine

The quantification results were presented in Fig. 1K. The term "block" used in the text referred to the inhibitory effect of quinidine on IKNa.

New comment:The fact remains that the term "block" is too strong for an effect that is incomplete. Also, the authors should add to the paper that Li+ is a weaker activator, so the reader knows some of the caveats to the approach.

Answer: We thank the reviewer for raising this point. We have added related citations and replaced the term “block” with “inhibit” in the revised manuscript.

Original:4. In K, for the WT, why is the effect of quinidine only striking for the largest currents?

Answer: We thank the reviewer for raising this point. After conducting an analysis, we found no correlation between the inhibitory effect of quinidine and the amplitudes of baseline IKNa in WT neurons (p = 0.6294) (Author response image 3). Therefore, the effect of quinidine is not solely limited to targeting the larger currents.

**Author response image 3. sa4fig3:** The correlation between the inhibitory effect of quinidine and the amplitudes of baseline IKNa in WT neurons (data from manuscript Figure 1K). r = 0.1555, p=0.6294, Pearson correlation analysis.

New comment:Please add this to the paper and the figure as Supplemental.

Answer: We thank the reviewer for raising this point. We have added this figure as Fig.S3B in the revised manuscript.

Original:1. Figure 2 A. The argument could be better made if the same concentration of quinidine were used for Slack and Slack + Nav1.6. It is recognized a greater sensitivity to quinidine is to be shown but as presented the figure is a bit confusing."

Answer: We apologize for the confusion. We would like to clarify that the presented concentrations of quinidine were chosen to be near the IC50 values for Slack and Slack+NaV1.6.

New comment:Please add this to the paper.

Answer: We thank the reviewer for raising this point. We have added the clarification about the presented concentrations in the revised manuscript.

Original:2C. Can the authors add the effect of quinidine to the condition where the prepulse potential was 90?"

Answer: We apologize for the confusion. We would like to clarify that the condition of prepulse potential at -90 mV is the same as the condition in Fig. 1. We only changed one experiment condition where the prepulse potential was changed to -40 mV from -90 mV.

New comment:There was no confusion. The authors should consider adding the condition where the prepulse potential was -90.

Answer: We thank the reviewer for raising this point. We have added the clarification about the voltage condition in the revised manuscript (see in Fig. 2A caption).

Original:2A. Clarify these 6 panels.

Answer: We thank the reviewer for raising this point. We have clarified the captions of Fig. 3A in the revised manuscript.

New comment: Clarification is needed. What is the blue? DAPI? What area of hippocamps? Please label cell layers. What area of cortex? Please label layers.

Answer: We thank the reviewer for raising this point. We have included the clarification in the Figure caption.

Original:Figure 7. The images need more clarity. They are very hard to see. Text is also hard to see.

Answer: We apologize for the lack of clarity in the images and text. we would like to provide a concise summary of the key findings shown in this figure.

Figure 7 illustrates an innovative intervention for treating SlackG269S-induced seizures in mice by disrupting the Slack-NaV1.6 interaction. Our results showed that blocking NaV1.6-mediated sodium influx significantly reduced Slack current amplitudes (Fig. 2D,G), suggesting that the Slack-NaV1.6 interaction contributes to the current amplitudes of epilepsy-related Slack mutant variants, aggravating the gain-of-function phenotype. Additionally, Slack’s C-terminus is involved in the Slack-NaV1.6 interaction (Fig. 5D). We assumed that overexpressing Slack’s C-terminus can disrupt the Slack-NaV1.6 interaction (compete with Slack) and thereby encounter the current amplitudes of epilepsy-related Slack mutant variants.

In HEK293 cells, overexpression of Slack’s C-terminus indeed significantly reduced the current amplitudes of epilepsy-related SlackG288S and SlackR398Q upon co-expression with NaV1.5/6NC (Fig. 7A,B). Subsequently, we evaluated this intervention in an in vivo epilepsy model by introducing the Slack G269S variant into C57BL/6N mice using AAV injection, mimicking the human Slack mutation G288S that we previously identified (Fig. 7C-G).

New comment:The images do not appear to have changed. Consider moving labels above the images so they can be distinguished better. Please label cell layers. Consider adding arrows to the point in the figure the authors want the reader to notice. The study design and timeline are unclear. What is (1) + (3), (2), etc.?

Answer: We thank the reviewer for pointing this out. We have modified Figure 7 in the revised manuscript and included the cell layer information in the Figure caption.

Original:It is not clear how data were obtained because injection of kainic acid does not lead to a convulsive seizure every 10 min for several hours, which is what appears to be shown. Individual seizures are just at the beginning and then they merge at the start of status epilepticus. After the onset of status epilepticus the animals twitch, have varied movements, sometime rear and fall, but there is not a return to normal behavior. Therefore one can not call them individual seizures. In some strains of mice, however, individual convulsive seizures do occur (even if the EEG shows status epilepticus is occurring) but there are rarely more than 5 over several hours and the graph has many more. Please explain."

Answer: We apologize for the confusion. Regarding the data acquisition in relation to kainic acid injection, we initiated the timing following intraperitoneal injection of kainic acid and recorded the seizure scores of per mouse at ten-minute intervals, following the methodology described in previous studies4.

Huang Z, Walker MC, Shah MM. Loss of dendritic HCN1 subunits enhances cortical excitability and epileptogenesis. J Neurosci. Sep 2 2009;29(35):10979-88. doi:10.1523/JNEUROSCI.1531-09.2009

The seizure scores were determined using a modified Racine, Pinal, and Rovner scale5,6: (1) Facial movements; (2) head nodding; (3) forelimb clonus; (4) dorsal extension (rearing); (5) Loss of balance and falling; (6) Repeated rearing and failing; (7) Violent jumping and running; (8) Stage 7 with periods of tonus; (9) Dead.

Pinel JP, Rovner LI. Electrode placement and kindling-induced experimental epilepsy. Exp Neurol. Jan 15 1978;58(2):335-46. doi:10.1016/0014-4886(78)90145-0Racine RJ. Modification of seizure activity by electrical stimulation. II. Motor seizure. Electroencephalogr Clin Neurophysiol. Mar 1972;32(3):281-94. doi:10.1016/00134694(72)90177-0

New comment:This was clear. Perhaps my question was not clear. The question is how one can count individual seizures if animals have continuous seizures. It seems like the authors did not consider or observe status epilepticus but individual seizures. If that is true the data are hard to believe because too many seizures were counted. Animals do not have nearly this many seizures after kainic acid.

Answer: We appreciate the reviewer’s clarification. Our methodology involved assessing the maximum seizure scale during 10-minute intervals per mouse as previously described7, rather than counting individual seizures. For instance, a mouse exhibited the loss of balance and falling multiple times within 30-40 minute interval, we recorded the seizure scale as 5 for that time interval.

1. Kim EC, Zhang J, Tang AY, et al. Spontaneous seizure and memory loss in mice expressing an epileptic encephalopathy variant in the calmodulin-binding domain of Kv7.2. Proc Natl Acad Sci U S A. Dec 21 2021;118(51)doi:10.1073/pnas.2021265118

**Reviewer #3 (Recommendations For The Authors):**
While the authors have improved the manuscript, several outstanding issues still need to be addressed. Some may have been missed because there is no response at all, but others may have been unclear.

Answer: We thank the reviewer for the criticisms. We have added additional experimental data and analysis to address the concerns.

Original issue from Public Review:1. Immunolabeling of the hippocampus CA1 suggests sodium channels as well as Slack colocalization with AnkG (Fig 3A). Proximity ligation assay for NaV1.6 and Slack or a super-resolution microscopy approach would be needed to increase confidence in the presented colocalization results. Furthermore, coimmunoprecipitation studies on the membrane fraction would bolster the functional relevance of NaV1.6-Slack interaction on the cell surface.

Answer: We thank the reviewer for good suggestions. We acknowledge that employing proximity ligation assay and high-resolution techniques would significantly enhance our understanding of the localization of the Slack-NaV1.6 coupling.

At present, the technical capabilities available in our laboratory and institution do not support highresolution testing. However, we are enthusiastic about exploring potential collaborations to address these questions in the future. Furthermore, we fully recognize the importance of conducting coimmunoprecipitation (Co-IP) assays from membrane fractions. While we have already completed Co-IP assays for total protein and quantified the FRET efficiency values between Slack and NaV1.6 in the membrane region, the Co-IP assays on membrane fractions will be conducted in our future investigations.

New comment from reviewer: so far, the authors have not demonstrated that Nav1.6 and Slack interact on the cell surface.

Answer: We thank the reviewer for pointing this out. We acknowledgement that our data did not directly demonstrate interaction between NaV1.6 and Slack on the cell surface and we have removed related terminology in the revised manuscript. Notably, our patch-clamp experiments in Fig. 2D,G and Fig. S10B showed a Na+-mediated membrane current coupling of Slack and NaV1.6. Additionally, the FRET efficiency values between Slack and NaV1.6 were quantified in the membrane region. These findings suggest that membrane-near Slack interacts with NaV1.6.

2. Although hippocampal slices from Scn8a+/- were used for studies in Fig. S8, it is not clear whether Scn8a-/- or Scn8a+/- tissue was used in other studies (Fig 1J & 1K). It will be important to clarify whether genetic manipulation of NaV1.6 expression (Fig. 1K) has an impact on sodiumactivated potassium current, level of surface Slack expression, or that of NaV1.6 near Slack.

Answer: We thank the reviewer for pointing this out. In Fig. 1G,J,K, primary cortical neurons from homozygous NaV1.6 knockout (Scn8a-/-) mice were used. We will clarify this information in the revised manuscript. In terms of the effects of genetic manipulation of NaV1.6 expression on IKNa and surface Slack expression, we compared the amplitudes of IKNa measured from homozygous NaV1.6 knockout (NaV1.6-KO) neurons and wild-type (WT) neurons. The results showed that homozygous knockout of NaV1.6 does not alter the amplitudes of IKNa (Author response image 4). The level of surface Slack expression will be tested further.

**Author response image 4. sa4fig4:** The amplitudes of IKNa in WT and NaV1.6-KO neurons (data from manuscript Figure 1K). ns, p > 0.05, unpaired two-tailed Student’s t test.

New comment from reviewer: The current version of the manuscrip>t does not contain these pertinent details and needs to be updated to include the information pertaining homozygous NaV1.6 knockouts. What age were these homozygous NaV1.6 knockout mice? These details need to be clearly stated in the manuscript.

Answer: We thank the reviewer for pointing this out. We have included this analysis in the revised manuscript (see Fig. S3A). The age of homozygous NaV1.6 knockout mice are P0-P1 and we have added this detail in the revised manuscript.

3, Did the epilepsy-related Slack mutations have an impact on NaV1.6-mediated sodium current?

Answer: We thank the reviewer’s question. We examined the amplitudes of NaV1.6 sodium current upon expression alone or co-expression of NaV1.6 with epilepsy-related Slack mutations (K629N, R950Q, K985N). The results showed that the tested epilepsy-related Slack mutations do not alter the amplitudes of NaV1.6 sodium current (Author response image 5).

**Author response image 5. sa4fig5:** The amplitudes of NaV1.6 sodium currents upon co-expression of NaV1.6 with epilepsy-related Slack mutant variants (SlackK629N, SlackR950Q, and SlackK985N). ns, p>0.05, oneway ANOVA followed by Bonferroni’s post hoc test.

New comment from reviewer: Figure with the functional effect of co-expression of NaV1.6 with epilepsy-related Slack mutations should be included in the revised manuscript

Answer: We thank the reviewer for pointing this out. We have included this analysis in the revised manuscript (see Fig. S10A).

Original issue from Recommendations For The Authors:1. A reference to homozygous knockout is made in the abstract; however, only heterozygous mice are mentioned in the methods section. The genotype of the mice needs to be made clear in the manuscript. Furthermore, at what age were these mice used in the study. Since homozygous knockout of NaV1.6 is lethal at a very young age (<4 wks), it would be important to clarify that point as well.

Answer: We thank the reviewer for pointing this out. In the revised manuscript, we have included information about the source of the primary cortical neurons used in our study. These neurons were obtained from postnatal homozygous NaV1.6 knockout C3HeB/FeJ mice and their wild-type littermate controls.

New comment from reviewer: The answer that postnatal homozygous NaV1.6 knockout C3HeB/FeJ mice were used is insufficient. What age were these mice? This needs to be clearly stated in the manuscript.

Answer: We thank the reviewer for pointing this out. The postnatal homozygous NaV1.6 knockout C3HeB/FeJ mice and their wild-type littermate controls are in P0-P1. We have included this information in the revised manuscript.

2. How long were the cells exposed to quinidine before the functional measurement were performed?

Answer: We thank the reviewer for pointing this out. The cells were exposed to the bath solution with quinidine for about one minute before applying step pulses.

New comment from reviewer: This needs to be clearly stated in the manuscript.

Answer: We thank the reviewer for pointing this out. We have included this information in the revised manuscript (see in Methods section).

3. In Fig. 6B-D, it is not clear to what extent co-expression of Slack mutants and NaV1.6 increases sodium-activated potassium current.

Answer: We thank the reviewer for pointing this out. We notice that the current amplitudes of Slack mutants exhibit a considerable degree of variation, ranging from less than 1 nA to over 20 nA (n = 58). To accurately measure the effects of NaV1.6 on increasing current amplitudes of Slack mutants, we plan to apply tetrodotoxin in the bath solution to block NaV1.6 sodium currents upon coexpression of Slack mutants with NaV1.6.

New comment from reviewer: Were these experiments with TTX completed? If so, they should be added to the revised manuscript.

Answer: We thank the reviewer for pointing this out. We compared the current amplitudes of epilepsy-related Slack mutant (SlackR950Q) before and after bath-application of 100 nM TTX upon co-expression with NaV1.6 in HEK293 cells. The results showed that bath-application of TTX significantly reduced the current amplitudes of SlackR950Q at +100 mV by nearly 40% (Author response image 6), suggesting NaV1.6 contributes to the current amplitudes of SlackR950Q. We have included this data in the revised manuscript (see Fig. S10B).

**Author response image 6. sa4fig6:** The current amplitudes of SlackR950Q before and after bath-application of 100 nM TTX upon co-expression with NaV1.6 in HEK293 cells (n = 5). ***p < 0.001, Two-way repeated measures ANOVA followed by Bonferroni’s post hoc test.

Additionally, we have corrected some errors in the methods and figure captions section:

Line 513, bath solution “5 glucose” should be “10 glucose.”Figure 3A caption, the description “hippocampus CA1 (left) and neocortex (right)” was flipped and we have corrected it.

References

1. Zhang Z, Rosenhouse-Dantsker A, Tang QY, Noskov S, Logothetis DE. The RCK2 domain uses a coordination site present in Kir channels to confer sodium sensitivity to Slo2.2 channels. J Neurosci. Jun 2 2010;30(22):7554-62. doi:10.1523/JNEUROSCI.0525-10.2010

2. Kaczmarek LK. Slack, Slick and Sodium-Activated Potassium Channels. ISRN Neurosci. Apr 18 2013;2013(2013)doi:10.1155/2013/354262

3. Budelli G, Hage TA, Wei A, et al. Na+-activated K+ channels express a large delayed outward current in neurons during normal physiology. Nat Neurosci. Jun 2009;12(6):745-50. doi:10.1038/nn.2313

4. Huang Z, Walker MC, Shah MM. Loss of dendritic HCN1 subunits enhances cortical excitability and epileptogenesis. J Neurosci. Sep 2 2009;29(35):10979-88.doi:10.1523/JNEUROSCI.1531-09.2009

5. Pinel JP, Rovner LI. Electrode placement and kindling-induced experimental epilepsy. Exp Neurol. Jan 15 1978;58(2):335-46. doi:10.1016/0014-4886(78)90145-0

6. Racine RJ. Modification of seizure activity by electrical stimulation. II. Motor seizure.Electroencephalogr Clin Neurophysiol. Mar 1972;32(3):281-94. doi:10.1016/0013-4694(72)90177-0

7. Kim EC, Zhang J, Tang AY, et al. Spontaneous seizure and memory loss in mice expressing an epileptic encephalopathy variant in the calmodulin-binding domain of Kv7.2. Proc Natl Acad Sci U S A. Dec 21 2021;118(51)doi:10.1073/pnas.2021265118